# BRAIN-INFORMED LANGUAGE MODEL TRAINING ENABLES SCALABLE AND GENERALIZABLE ALIGNMENT WITH HUMAN BRAIN ACTIVITY

## ABSTRACT

Language models (LM) provide rich representational spaces that partially align with neural activity during naturalistic experiences such as movie watching. Yet, leveraging brain recordings to actively guide LM training remains underexplored. Here, we address this question by investigating whether functional MRI (fMRI) recordings can guide language model training by aligning language representations with brain dynamics. Using over 50 hours of fMRI data from six subjects watching Friends, plus 10 hours of held-out movies, we augment both pre-trained and randomly initialized language models with a brain-alignment module and compare three regimes: (i) a brain-fine-tuned LM that updates a small set of adapter parameters on top of a pretrained model, (ii) a brain-from-scratch LM trained only from brain supervision, and (iii) a frozen text-only baseline with a fully connected brain encoding layer. Our results show three main findings. First, brain-informed fine-tuning consistently outperforms text-only baselines and brain-from-scratch models, with voxel-level gains that scale with both model size (GPT-2 124M and LLaMA-2 7B), eliciting robust cross-subject and cross-stimulus encoding. Second, a dual-objective loss that balances language modeling with brain-alignment surpasses a brain-only loss that optimizes only for fMRI prediction, producing more stable and generalizable encoders. Finally, brain supervision enriches LM representations with multisensory inductive biases: brain-fine-tuned models outperform unimodal baselines on VL-Commonsense, better capturing perceptual and associative properties (e.g., color, shape, co-occurrence) that text-only training underrepresents. Together, these results establish cortical dynamics as an effective supervisory signal, enabling scalable, generalizable, and brain-aligned LMs that internalize aspects of human-like multimodal representation.

## 1 INTRODUCTION

Recent research has demonstrated that language models (LMs) provide rich representations that can predict brain activity during naturalistic experiences such as listening to stories or watching movies Jain & Huth (2018); Toneva & Wehbe (2019); Caucheteux & King (2022); Schrimpf et al. (2021); Toneva et al. (2022). In most studies, however, brain activity serves only as a dependent variable: models explain neural responses but are not themselves shaped by them. Using brain recordings as a training signal to guide LMs remains comparatively rare Schwartz et al. (2019), leaving open whether biologically grounded supervision can improve alignment with brain activity or enrich text-trained representations with the multimodal information to extend their unimodal representations toward more human-like ones?

Despite its promise, leveraging brain data to train LMs poses unique challenges. While modern functional magnetic resonance imaging (fMRI) datasets provide high-quality measurements of brain activity during naturalistic language comprehension, brain signals differ fundamentally from textual corpora: fMRI responses are high-dimensional, slow, and spatially structured. Yet they carry rich information that is not fully reflected in text content that would capture complex cognitive processes like mental imagery, social reasoning, and predictive inference. Developing models that can effectively learn from these complex brain patterns, without relying solely on conventional linguistic supervision, remains an open frontier. This motivates several key questions: Can brain supervision

improve the LM-brain alignment? If this holds, how does fine-tuning a pretrained LM compare to training an LM solely with brain data? And how do such models scale with data quantity, model size, and subject variability?

To address these questions, we build on a large-scale naturalistic fMRI dataset from the CNeuroMod project Boyle et al. (2020), comprising more than 50 hours of *Friends* and 10 hours of diverse movie stimuli watching. We systematically explore three model regimes that share a common architecture combining an LM backbone, a hemodynamic convolution layer, and a voxelwise linear encoding: (1) *brain-fine-tuned*, where a pretrained LM is adapted using LoRA modules and a linear readout; (2) *brain-from-scratch*, trained only with brain supervision; (3) *text-only* baseline, where the LM is frozen and only a linear mapping to voxels is learned. We further evaluate two objectives: a brain-only loss and a dual-loss combining brain prediction with language modeling.

We test three core hypotheses. First, the brain-fine-tuned model should aid the emergence of brain-aligned representations, outperforming text-only and brain-from-scratch models, as pre-trained linguistic structure provides a complementary substrate for learning brain-model alignment. Second, alignment should scale with both LM capacity and the amount of brain data. Third, because naturalistic audiovisual experience is richly multimodal, brain supervision may endow the model with sensory and associative information that text training alone underrepresents, improving visually grounded commonsense reasoning.

Our findings support all three hypotheses. Brain-fine-tuned models reliably outperform text-only baselines across cortical regions, including temporal and frontal areas, with larger LMs yielding larger gains. Scaling training from 1-40 hours produces continued improvements and strong generalization to new subjects and unseen movies. A dual LM + brain objective stabilizes training and avoids representational collapse relative to brain-only optimization. Beyond encoding, brain-aligned models improve performance on visually grounded commonsense judgments, using the Visual Commonsense Test (ViComTe) Zhang et al. (2022), a benchmark probing sensory and associative attributes (e.g., color, shape, co-occurrence), suggesting that brain data collected on rich audiovisual stimuli provides enriched representations that are not captured by text only. At the cortical level, difference maps reveal that brain-informed training enhances encoding not only in the language network but also in areas associated with social inference and visual meaning Huth et al. (2016).

It is important to emphasize that, throughout this work, the aim of this work is not to improve the general linguistic competence of language models; rather, we treat pretrained LMs as representational backbones and ask whether brain-supervised adaptation yields better neural encoding and richer multimodal representations. Together, our results show that cortical dynamics provide an effective inductive bias for shaping LM representations, supporting cross-subject and out-of-distribution generalization and offering a path toward models that more closely reflect the structure of naturalistic human cognition. Our approach resembles a novel form of knowledge distillation, in which the LM learns internal processing structures through alignment with human brain activity rather than from a larger teacher model.

## 2 RELATED WORK

Learning to align linguistic representations with brain activity has been a long-standing goal in the neuroscience of language. Early work showed that distributional semantic models map reliably onto neural activation patterns during language comprehension Fyshe et al. (2014). More recent studies have attempted fine-tune pretrained LMs using neuroimaging data. For instance, Schwartz et al. (2019) fine-tuned BERT (Devlin et al., 2018) on approximately one hour of fMRI And magnetoencephalography (MEG)) per participant, reporting improved brain prediction and generalization across subject and imaging modalities, though without evidence of improvement in downstream task. Similarly, Negi et al. (2025) showed that fine-tuning monolingual and multilingual LMs on bilingual fMRI data can enhance both neural encoding and linguistic performance. Although their dataset was much smaller than ours (roughly 2 hours per subject), and relied on multilanguage contrasts rather than naturalistic audiovisual stimulation, their results suggest that neuroimaging data can acts as a useful multimodal training signal for LMs.

Related work in speech modeling has explored neuro-supervised adaptation in architectures tailored to spoken language. Moussa et al. (2024) fine-tuned a speech-LM on brain responses to auditory narratives and reported gains on semantic tasks, though, they did not look at benefits of fine-tuning

regular LMs. In parallel, Vattikonda et al. (2025) introduced BrainWavLM, adding an auxiliary brain prediction loss to a wav2vec2 which improved speech-brain alignment. These studies collectively indicate that bran data can guide model representations, but they focus on comparatively small datasets and speech-based architectures (Schwartz et al., 2019; Negi et al., 2025; Moussa et al., 2024; Vattikonda et al., 2025). In contrast, our work examines brain-supervised adaptation of general-purpose text LMs under naturalistic audiovisual stimulation, systematically evaluating multiple brain alignment regimes, cross-dataset generalization, and the effects of scaling to substantially larger quantities of neural data.

A complementary line of research has explored improving neural predictability without directly training on brain data. Structural modifications, such as manipulating attention patterns in upper layers Toneva & Wehbe (2019) showed that removing attention from the upper layers of a transformer improved brain activity prediction as well as performance on syntactic tasks. This finding suggested overlapping representational hierarchies between LMs and neural processing. Aw & Toneva (2022) demonstrated that fine-tuning an LM on a narrative dataset enhanced its ability to predict fMRI responses. Zhou et al. (2024) further reported that LMs underrepresent certain cognitive dimensions, such as emotions or physical reasoning and that targeted fine-tuning on relevant tasks can improve MEG-based predictability.

In parallel, several studies have systematically evaluated how well large pretrained LMs predict brain activity without brain-based supervision. Caucheteux & King (2022) demonstrated partial convergence between the activations of large LMs and brain responses during natural language processing, while Schrimpf et al. (2021) evaluated a wide range of architectures to benchmark brain prediction across the brain. These efforts reveal substantial intrinsic overlap between LM activations and cortical responses but do not investigate whether incorporating neural supervision can further strengthen or restructure this alignment.

## 3 METHODS

### 3.1 BRAIN DATA

We used a set of fMRI data, as part of CNeuroMod naturalistic movie-viewing datasets (Boyle et al., 2020), which were acquired from six English-speaking participants (three women) using a 3T Siemens PrismaFit scanner. The dataset comprises fMRI data of watching 50 hours of *Friends* TV show across six seasons (see Figure 1a) and 10 hours of brain recording of the Movie10 dataset, including *The Wolf of Wall Street, The Bourne Supremacy, Hidden Figures, and Life*. Each Friends episode and movie was divided into segments of around 12 minutes to eliminate the fatigue of watching. Participants watched segments sequentially, though not necessarily in a single session. The details of the fMRI data processing could be found in Appendix A.1.

Season 3 of Friends was held out as an unseen test set across all experiments. The remaining seasons were used for training and validation, following an incremental training scheme with datasets of 1, 5, 10, 20, 30, and 40 hours. For each training data size, 10% of the data was reserved for validation. Movie segments were randomly shuffled for both training and validation datasets. Additionally, the Movie10 dataset[1] was used to evaluate model performance on out-of-sample data.

### 3.2 TEXT INPUT AND TRANSCRIPT PROCESSING

The main input to the model consisted of the tokenized, HRF-weighted movie transcripts. We first extracted the transcripts for the *Friends* and *Movie10* from the original movie DVDs and aligned them to the audio track using a custom pipeline built on AssemblyAI, Inc. (2025)'s automatic speech-to-text pipeline, which provided word-level timestamps. Each word was then tokenized using the corresponding LM tokenizer, and all resulting sub-tokens inherited the timestamp of the word from which they were generated. For each fMRI TR, we selected the 32 most recent tokens preceding that TR and applied a canonical HRF to compute token-specific weights reflecting their expected contribution to the measured BOLD response. Additional details about the HRF convolution procedure and the choice of the temporal context window are provided in section 3.4.

---

[1]A complementary naturalistic movie fMRI dataset used for generalization testing `https://docs.cneuromod.ca/en/latest/DATASETS.html#movie10`.

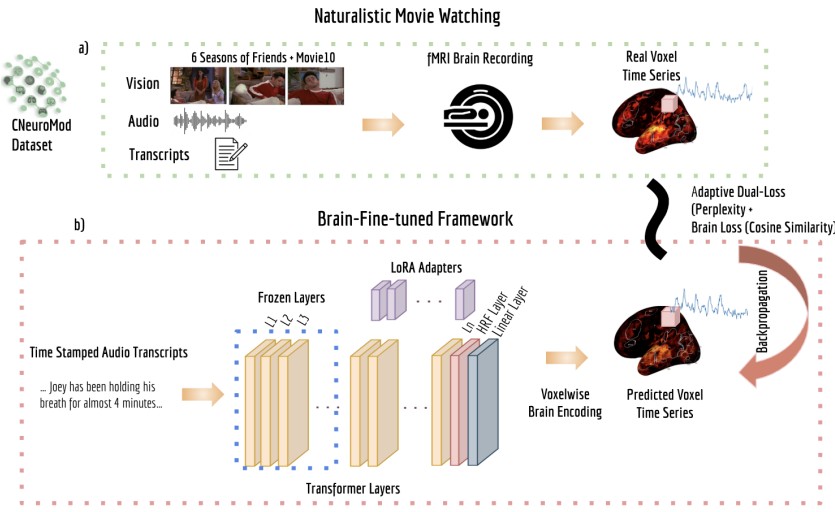

Figure 1: End-to-end approach for aligning language model representations with brain activity measured during naturalistic viewing. a) Participants watched Friends episodes while undergoing fMRI. Movie audio was transcribed and time-aligned, providing word onsets that serve as text inputs to the model. fMRI measures the BOLD signal (blood–oxygen–level–dependent activity) every TR (repetition time) at each voxel. (b) At each TR, the model receives the most recent transcript tokens, weighted by a canonical HRF (hemodynamic response function) to model the delayed BOLD response. These tokens pass through a pretrained Transformer LM (GPT-2 or LLaMA-2-7B), where only selected upper layers are adapted using LoRA for parameter-efficient fine-tuning. A linear voxelwise head predicts fMRI activity, optimized with a dual loss (language perplexity + brain-prediction loss). We compare brain-fine-tuned and brain-from-scratch training regimes.

## 3.3 VISUAL COMMONSENSE DATASET

We evaluated brain-fine-tuned models on VL-Commonsense (CoDa) Zhang et al. (2022), to assess whether brain supervision enriches unimodal LMs with sensory and associative knowledge that is implicit in neural signals but difficult to learn from text alone, which text-only LMs often under-represent. The dataset comprises a set of nouns and probes visually grounded properties of every-day objects (e.g., color, shape, material, size, and co-occurrence). Following standard prompting, we scored candidate attributes for each noun-relation pair using log-probabilities, with a Pointwise Mutual Information (PMI) de-biasing and tokenization controls, and reported accuracy and rank correlation against human judgments. The details about the evaluation of the CoDa dataset could be found in Appendix A.2.

## 3.4 MODEL ARCHITECTURE

We implemented a brain-aligned LM under two distinct training regimes:

• *Brain-fine-tuned* Model: A pretrained LM backbone is partially adapted using LoRA modules inserted into the upper transformer layers, while all other weights remain fixed. Both a *brain-only* objective and a *dual-loss* objective are evaluated (the latter described below).

• *Brain-from-scratch* Model: The LM backbone is randomly initialized and trained end-to-end using only the fMRI experiment. This model serves as a conceptual ablation to test whether meaningful structure can emerge from brain supervision alone. Removing all pretrained linguistic knowledge isolates the contribution of brain data and highlights the contribution of the pretrained representations in the brain-fine-tuned model. Throughout the paper, we treat this as a lower-bound control rather than a practical modeling strategy.

• *Text-only Baseline*: The LM backbone is frozen, and only the linear encoding layer is trained. This model serves as a non-fine-tuned baseline for representational alignment.

All models share a common pre-trained transformer-based LM backbone, either GPT-2(124M) or LLaMA2(7B), augmented with a hemodynamic response function (HRF) layer for temporal align-

ment and a fully connected voxelwise encoding layer to align latent representations with voxelwise fMRI data. This end-to-end framework allows us to examine how neural supervision modifies or enriches the representational geometry of pretrained LMs.

Since our main goal is to evaluate the shared representational space learned by LM, independent of spatial priors or anatomical smoothing, we adopt voxelwise linear encoding to predict fMRI responses from the LM features. As discussed in Appendix A.3, this choice avoids imposing additional inductive biases on spatial structure. For across-subject generalization, voxel coordinates and response profiles vary across individuals, so we re-fit a subject-specific Ridge regression layer while keeping the LM representations fixed. This approach preserves the learned shared representational space while allowing a simple linear mapping to adapt to each participant's voxel geometry (Appendix A.4).

**Parameter-Efficient Adaptation with LoRA**   In the brain-fine-tuned model, we use Low-Rank Adaptation (LoRA) Hu et al. (2022), a parameter-efficient fine-tuning method that inserts trainable low-rank matrices into selected upper LM layers. LoRA is particularly suitable for neuroimaging: (i) it enables tractable adaptation of large pretrained LMs to comparatively small fMRI datasets, and (ii) it constrains updates to a low-rank subspace, preserving the backbone's pretrained linguistic representations while still allowing the model to integrate neural constraints. This balance of efficiency and representational stability makes LoRA a natural choice for brain alignment.

**HRF Convolution and Temporal Context**   Because fMRI responses are temporally delayed and sluggish, we follow standard practice in naturalistic fMRI encoding (e.g., Huth et al. 2016; Caucheteux & King 2022) and convolve token onset times with a canonical single-gamma HRF:

$$\mathrm{HRF}(t) = \begin{cases} t^{8.6}\, e^{-t/0.547}, & t \le 15 \\ 0, & t > 15, \end{cases}$$

where $t$ denotes the temporal offset (in seconds) between a token and the center of TR being predicted (see Supplementary Figure S3). This yields a biologically motivated temporal integration window of roughly 12-15 seconds and is applied as a weight over the 32 most recent tokens preceding each TR (1.49 s). The 32-token window corresponds to 5-6 seconds of preceding speech under canonical HRF and was selected after pilot sweeps over 16-64 tokens, which gave comparable results. HRF weights are applied directly to the final-layer LM hidden states, producing a weighted linear combination approximating standard HRF convolution across feature dimensions(Figure 1b).

**Dual-Loss Objective**   Training was run end-to-end: HRF-weighted LM representations feed into the encoding layer to predict voxelwise BOLD responses. To maintain training stability under fMRI-supervised adaptation, we use an adaptively weighted dual-loss objective combining voxel prediction and language modeling loss:

$$\mathcal{L} = w_{\mathrm{brain}}\, \mathcal{L}_{\mathrm{brain}} + w_{\mathrm{lm}}\, \mathcal{L}_{\mathrm{LM}} + \lambda \left\| W_{\mathrm{ridge}} \right\|_2^2.$$

This follows established findings in multitask learning and continual adaptation (Caruana, 1997; Goodfellow et al., 2014; Aghajanyan et al., 2021), where retaining a strong pretrained objective stabilizes optimization when the new supervision signal is comparatively small. Here, the LM loss acts as a conservative anchor that preserves the backbone's linguistic structure, while LoRA updates incorporate brain-derived constraints. Our goal is not to improve general-purpose language modeling; the LM loss serves purely as a stabilizing mechanism. The adaptive weighting schedule is described in Appendix A.5.

**Hyperparameter Optimization**   We performed systematic sweeps over learning rate, weight decay, ridge regularization, and dropout (Appendix A.9 and A.10). Dropout consistently reduced stability and performance, whereas learning rate and weight decay had the strongest effects. For LoRA, intermediate ranks (8 for GPT-2 and 16 for LLaMA-2) offered the best balance between efficiency and predictive accuracy. Models were optimized with AdamW using separate parameter groups and fixed learning rates, selected from the sweep results.


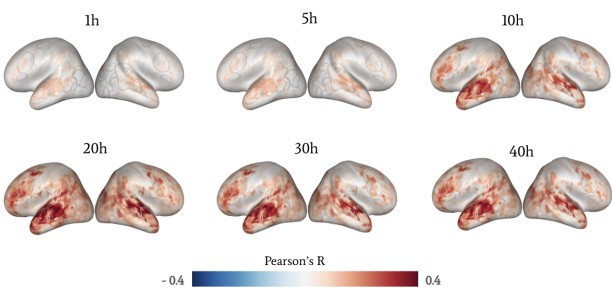

Figure 2: Effect of training data volume on brain–LM alignment for GPT-2. Voxelwise prediction performance (Pearson's r) increases systematically with the amount of brain data used during fine-tuning, shown here for models trained on 1h, 5h, 10h, 20h, 30h, and 40h of naturalistic movie-watching fMRI. Larger training sets yield broader and stronger cortical alignment, especially in temporal, parietal, and occipitotemporal regions, demonstrating that neural supervision continues to scale with data and does not saturate appat small sample sizes. Color intensity reflects voxelwise correlation between predicted and measured BOLD responses.

## 4 RESULTS

### 4.1 BRAIN ENCODING IMPROVES WITH BRAIN FINE-TUNING AND SCALES WITH BIGGER DATA AND MODEL SIZE

We first tested whether fine-tuning language models with brain activity improves their ability to predict neural responses. Figure 2 shows voxelwise performance on a held-out *Friends Season 3* test set for a brain-fine-tuned GPT-2 model trained on increasing amounts of fMRI data (1–40 hours) for one subject, and the remaining subjects are shown in S7. Encoding performance improves steadily with more training data, with marked gains emerging around 10 hours and continuing through 20 hours, where a plateau is reached. We use a bootstrap procedure to evaluate improvement at different voxels, extract p-value (see Appendix A.10), and correct for multiple comparisons at a false discovery rate of 0.05 (see Appendix Figure S6). Initial gains from 1 to 5 hours are focused in the auditory cortex, but after going to larger scales, these improvements become broadly distributed across bilateral temporal, frontal, and occipitotemporal cortices, indicating that brain-informed supervision enriches representations beyond regions tightly linked to auditory or linguistic input. Results for the brain-from-scratch model, which achieves notable gains in auditory and core language regions, and more limited gains in other regions, are provided in Appendix Figure S9.

Next, we asked whether scaling model size amplifies the benefits of brain fine-tuning. Figure 3 compares brain-fine-tuned GPT-2 (124M) and LLaMA-2 (7B). Both models capture widespread cortical responses, but the larger LLaMA consistently achieves higher accuracy. Difference maps highlight that fine-tuning LLaMA with brain data especially strengthens encoding in higher-order association cortices. Together, these results demonstrate that brain-fine-tuning benefits scale with both training data size and model capacity. Consistent with this, the training dynamics of the dual-loss objective (Appendix Fig. S8) show that LLaMA-2B can achieve stronger reductions in brain loss than GPT-2 at similar perplexity levels, although this advantage reflects both model capacity and a larger low-rank adaptation subspace.

### 4.2 BALANCING BRAIN AND LANGUAGE LOSSES YIELDS RICHER REPRESENTATIONS

We compared dual-loss and brain-only-loss training across increasing amounts of fMRI data (Fig. 4). At very short exposure (1h), brain-only-loss training achieves a higher mean correlation. However, from 5h onward, dual-loss consistently surpasses brain-only-loss, with the strongest advantage at 20h. At this point, dual-loss reaches its peak performance. The advantage persists through 30h, before slightly reversing at 40h, indicating possible overfitting.

A closer look to the dual optimization trajectories (Appendix Fig. S8) shows that early epochs are dominated by perplexity minimization, indicating that the model refines token-level linguistic representations before neural constraints takes effect. Once language improvements plateau, the scheduler

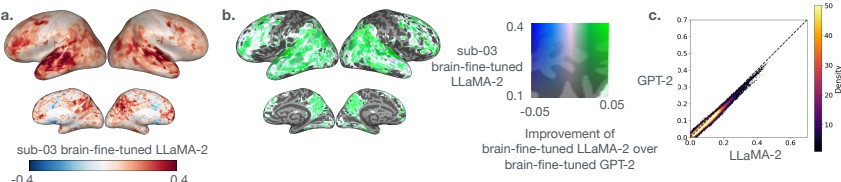

Figure 3: Comparison of brain-fine-tuned LLaMA-2 and GPT-2 in voxelwise prediction accuracy. a. Voxelwise Pearson correlation maps for the brain-fine-tuned LLaMA-2 model for a representative subject. b. Cortical map showing voxel-level improvements of brain-fine-tuned LLaMA-2 over brain-fine-tuned GPT-2, highlighting regions where the larger model yields the largest gains. c. Voxelwise scatter plot comparing the prediction performance of the two models; points above the diagonal indicate voxels better predicted by LLaMA-2. Overall, LLaMA-2 provides broad and reliable improvements over GPT-2 across the cortex.

shifts weight toward neural prediction, yielding continued reductions in brain loss without a collapse in linguistic competence. Although both model maintains stable language performance, LLaMA-2 exhibits a stronger reduction in brain loss than GPT-2 at a comparable perplexity, suggesting that larger models provide more representational capacity for neural alignment without inducing infer-ence with token-level modeling. Overall, these results show that the dual-loss schedule accelerates learning and yields higher encoding accuracy across most training regimes by reshaping the internal representations without collapsing them into a purely neural task solution.

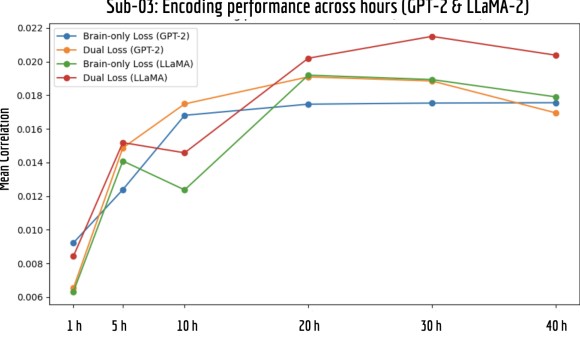

Figure 4: Encoding performance (mean voxelwise correlation) across training durations for sub-03 using GPT-2 and LLaMA backbones under brain-only-loss and dual-loss regimes. Note that performance is averaged across all voxels, leading to small magnitudes that don't reflect the scale of performance achieved in well-predicted regions. Dual-loss consistently outperforms brain-only-loss beyond 5h for both backbones, with the strongest gains observed for LLaMA at 20–30h. While GPT-2 plateaus earlier and benefits modestly from dual-loss, LLaMA continues to improve, highlighting that brain-informed supervision scales more effectively with larger models.

### 4.3 BALANCING BRAIN AND LANGUAGE LOSSES YIELDS RICHER REPRESENTATIONS

To further examine where neural supervision reshapes the model's internal representations, we com-pared voxelwise encoding performance across cortical regions grouped by functional domain (Lan-guage, Face/Body, Scene, Early Visual; see Appendix Fig. S4). Brain fine-tuning increased predic-tion accuracy across the majority of regions, with particularly strong improvements in frontal and temporal language areas (e.g., IFG, MFG), scene-selective cortex (MPA, OPA), and early visual areas (V1–V3). These effects indicate that neural supervision injects perceptual and multimodal structure into the model's representations, enriching them beyond what text-only pretraining affords. Overall, this suggests that brain alignment acts as a functional inductive bias that pushes language models toward representational geometries more consistent with human cortical organization.

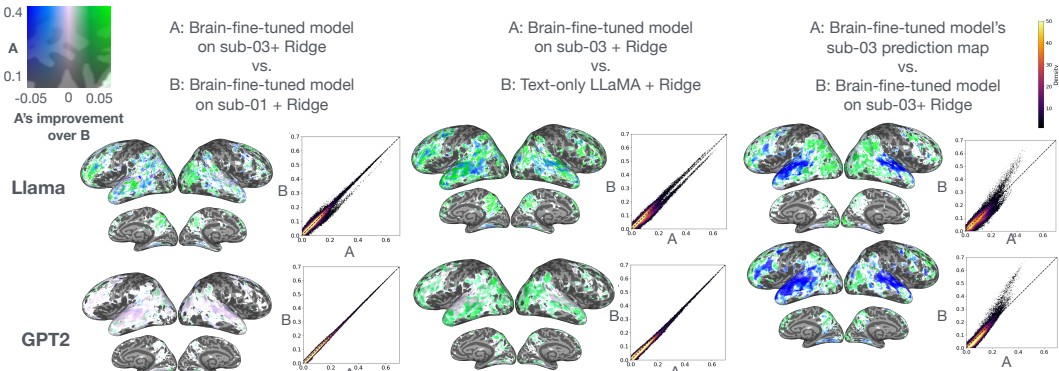

Figure 5: **Comparing representational features derived from brain-fine-tuned and baseline models across subjects and feature extraction stages.** Cortical maps and scatterplots showing voxelwise differences in prediction performance (Pearson's $r$) for pairs of models or feature sources. Positive values (green–cyan) indicate voxels better predicted by model A over model B, while negative values (blue) indicate the opposite. Left: Representations learned by a brain-fine-tuned model on sub-03 outperform those learned on sub-01 when evaluated on sub-03, demonstrating that subject-specific neural supervision produces representations that better align with individual cortical geometry. Middle: Brain-fine-tuned LLaMA features outperform text-only LLaMA features after Ridge readout, highlighting that neural supervision enriches the backbone's representational space beyond what is available from text-only pretraining. Right: The brain-fine-tuned prediction map exceeds the Ridge reconstruction in many regions, suggesting that joint optimization of the encoding head captures information not recoverable by a linear readout alone.

### 4.4 BRAIN-ALIGNED LMS GENERALIZE ACROSS SUBJECTS AT THE VOXEL LEVEL

To evaluate cross-subject generalization, we asked whether the representations learned from one subject's brain data could transfer to predict voxel responses in another subject. For each source subject, we froze all LM parameters after training and extracted the model's representations on the held-out stimuli for all other subjects. Because voxel coordinates and noise profiles differ across subjects, we trained a new voxelwise Ridge regression model for each test subject's fMRI responses. This re-fitting procedure of the Ridge model does not alter the learned LM representations, and it only estimates a linear mapping from a fixed shared representational space to each subject's voxel geometry, as a standard in encoding-model research where the representational space is shared, but the voxel readout must be subject specific (e.g., (Wehbe et al., 2015; Huth et al., 2016)).

Our results showed that encoding performance remained robust across all subject pairs (Appendix Figure S10), with consistent mean correlations across subjects and across training durations (1-40 hours) for both GPT-2 and LLaMA backbones. This indicates that the aligned signal extracted from one participant generalizes reliably to others and does not merely capture individual-specific idiosyncrasies. Figure 5 further shows that, for the best predicted voxels, primarily in temporal and frontal language-selective regions, a Ridge model trained on representations from a subject's own fine-tuned network outperforms the raw predictions produced directly by the fine-tuned encoding head. This reflects known behavior of voxelwise RidgeCV, which selects weaker regularization for high-SNR voxels and stronger shrinkage elsewhere (Wehbe et al., 2015). For lower-SNR voxels, the predictions from the fine-tuned encoding head perform better, suggesting complementary strengths between the jointly trained head and the post-hoc Ridge mapping. Together, these results show that brain-informed fine-tuning produces a shared representational space that is stable across individuals while allowing subject-specific linear encoders to exploit fine-grained voxel variability.

### 4.5 BRAIN-FINE-TUNED LMS GENERALIZE ROBUSTLY ACROSS NATURALISTIC STIMULI

We next assessed whether brain-informed training enables models to generalize to novel, out-of-distribution naturalistic stimuli. To this end, we evaluated the brain-fine-tuned models on the *Movie10* dataset, which includes audiovisual movies not present during training. As shown in Fig-

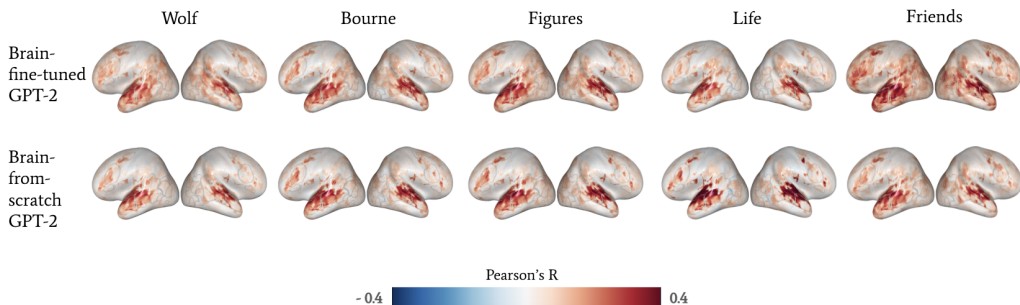

Figure 6: Cross-stimulus generalization of GPT-2 models trained with and without brain supervision. For each out-of-distribution movie (Wolf, Bourne, Figures, Life, Friends), we visualize voxelwise Pearson correlation between model-predicted and observed fMRI time series. The brain-fine-tuned GPT-2 (top) generalizes robustly, producing reliable predictions across large portions of the cortex despite never seeing these movies during training. In contrast, the brain-from-scratch GPT-2 (bottom) shows weaker and more localized accuracy, underscoring the importance of pretrained language structure for successful neural alignment.

ure 6, brain-fine-tuned models consistently predicted neural responses across all five unseen movies and in all subjects (see Appendix Figure S12), with widespread cortical activation patterns. This generalization held across higher-order association regions as well as sensory cortices, highlighting the robustness of the learned alignment.

| Model | Color (N=10,306) | Co-occur (N=2,108) | Material (N=6,176) | Shape (N=4,015) | Size (N=4,000) | Overall |
|---|---|---|---|---|---|---|
| GPT-2 (base) | 0.385 | 0.257 | 0.388 | 0.298 | 0.506 | 0.381 |
| GPT-2 (brain-finetuned) | 0.328 | 0.212 | 0.402 | 0.404 | 0.511 | 0.375 |
| LLaMA-2-7B (base) | 0.569 | 0.394 | 0.653 | 0.481 | 0.514 | 0.560 |
| LLaMA-2-7B (brain-finetuned) | 0.608 | 0.486 | 0.652 | 0.515 | 0.489 | 0.584 |

Table 1: VL-Commonsense (CoDa) results by relation (Color, Co-occur, Material, Shape, Size) for GPT-2 and LLaMA-2-7B, with and without brain fine-tuning. Metrics are top-1 accuracy (per relation; overall micro). Brain-fine-tuning consistently improves LLaMA, yielding gains especially on relations that depend on perceptual or associative knowledge (Co-occur, Color, Shape), while Material remains near the ceiling and Size shows minor variability. GPT-2 exhibits smaller or mixed changes. These patterns indicate that brain supervision might inject multisensory knowledge that text-only pretraining under-specifies.

### 4.6 BRAIN-FINETUNING ENRICHES SENSORY REPRESENTATIONS AND ENHANCES VISUALLY GROUNDED COMMONSENSE

We used VL-commonsense (CoDa), a text-based probe of visual–linguistic commonsense, to test whether brain-aligned finetuning, adapting an LM to predict neural responses to naturalistic *audio-visual* stimuli, improves a language model's grasp of visually grounded, multisensory regularities. The dataset comprised a set of nouns; for a given noun, the model must choose the most plausible color, material, shape, size, or typical co-occurring object from a candidate set. Although posed in text, these relations are grounded in regularities of the visual world (e.g., "banana-yellow", "mug-coffee"), which are underrepresented in purely textual corpora. Thus, observing gains on CoDA after neural supervision would suggest that fMRI signals inject perceptual structure not readily recoverable from text-only pretraining, yielding measurable gains on VL-commonsense judgments.

Across all relations in VL-Commonsense, expectedly, LLaMA-2-7B outperforms GPT-2, reflecting the advantage of scale and broader pretraining. More importantly, adapting LLaMA with brain supervision yields consistent improvements in some visually grounded commonsense. The largest absolute gain is in co-occurrence, followed by color and shape, attributes that depend on perceptual regularities and multisensory integration, and that are believed to be encoded in high-level cortical

regions. Other attributes (e.g., material, size) show smaller improvements. Material knowledge, already strong in the base LLaMA, remains flat. In contrast, GPT-2 shows smaller benefits.

These patterns suggest that the brain prediction objective encourages the LMs to encode sensory-grounded associations beyond what is available from text-only training. To corroborate the improvement in visual common sense, we noticed that the improvements in the fine-tuned model over the text-only baseline (Figure 5) were distributed in regions that are thought to represent social and visual information (Huth et al., 2016). While this observation remains to be tested in future work, it offers an interesting hypothesis that through multimodal exposure, the representation of the input text gets enriched with multimodal brain responses from non-language brain regions.

It should be noted that throughout the process of brain-fine-tuning, the models were trained only with the Friends transcripts, paired with the fMRI data were used. Thus, the improvements on the CoDa downstream task cannot be attributed to the additional linguistic exposure beyond the Friends transcripts. Overall, these results suggest that aligning LMs with the brain enables them to extract and represent sensory-rich, human-like associations beyond what unimodal text training affords.

## 5  DISCUSSION

Across our experiments, we find that incorporating brain data into model training yields more accurate, more generalizable, and more sensory-grounded representations than text-only training. These results position brain-informed fine-tuning as a step toward models that better capture the multidimensional nature of human cognition.

Our first major result is that fine-tuning substantially improves voxelwise encoding, with gains that continue to scale with additional neural data, beyond classical language regions into frontal, temporal, and occipitotemporal cortices, indicating that brain supervision encourages models to capture more holistic, multimodal representations. Second, larger backbones amplify the benefits: although both GPT-2 and LLaMA-2 improve, the larger model aligns more strongly with the cortical activity, particularly in association regions involved in semantic integration and mnemonic processing. This mirrors scaling trends in conventional LMs while grounding the improvements in biological representational structure. Third, brain-informed models generalize robustly across individuals and across stimuli. Representations learned from one participant transfer effectively to others, and models trained on Friends generalize to unseen movies. Finally, we show that brain supervision enriches language models with perceptual and associative knowledge that text-only pretraining underrepresents. On visually grounded commonsense judgments, brain-fine-tuned models outperform unimodal baselines. This result suggests that brain-informed fine-tuning can inject multisensory inductive biases into LMs, allowing them to represent aspects of the world that humans grasp intuitively but that corpus statistics alone often fail to convey.

Our approach has some limitations. fMRI has slow hemodynamics and limited temporal resolution. In our work, we used a model canonical HRF for each voxel for encoding; however, learning region-specific and voxel-specific hemodynamics could further improve fidelity. Although our dataset is deep-phenotyped, adapting LLMs to fMRI still risks overfitting; we mitigate this with LoRA, early stopping, and out-of-distribution evaluation. The brain-from-scratch model, while prone to overfitting, serves as a conceptual lower bound that clarifies the contribution of pretraining.

While we evaluate two model families and multiple data sizes, broader architectural sweeps would clarify how capacity interacts with neural supervision. We also adopt voxelwise linear to avoid spatial priors that complicate cross-subject comparisons; future work could explore spatially structured encoders that respect cortical geometry. Beyond CoDa, additional tasks probing social reasoning, event structure, or predictive inference may reveal which cognitive dimensions shift most under brain alignment.

Although our focus is on improving brain encoding, these findings suggest broader possibilities; brain-aligned models may form the basis of more human-grounded AI systems that integrate linguistic, perceptual, and cognitive structure. In the long term, such approaches may enable models that adapt both to shared and individual patterns of human brain activity.

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

## A  APPENDIX

### GLOSSARY OF TERMS

#### TRAINING REGIMES AND MODEL VARIANTS

- **Brain-fine-tuned Model:** A pre-trained LM augmented with LoRA adapters and fine-tuned on brain data; only LoRA modules and the encoding layer are trainable.
- **Brain-from-scratch Model:** A randomly initialized transformer LM trained solely on brain data.
- **Brain-only Loss:** A training objective consisting only of the brain-prediction loss; no language modeling term is used.
- **Text-only Baseline:** A model in which the LM backbone is kept frozen (fixed) and only a linear voxelwise encoding head is trained. Serves as a non-fine-tuned baseline.
- **Dual-loss Training (Brain + LM Loss):** An adaptively weighted objective combining brain-prediction loss and language-modeling loss.

- **Encoding Model:** A model mapping LM representations to voxelwise fMRI responses, via a linear regression head.

## NEUROIMAGING CONCEPTS

- **TR (Repetition Time):** Temporal resolution of fMRI acquisition; in this dataset, TR = 1.49s.
- **Voxel:** A 3D unit of fMRI measurement with a corresponding BOLD time series.
- **BOLD Signal:** Blood-oxygen-level dependent measurement serving as an indirect index of neural activity.
- **HRF (Hemodynamic Response Function):** Canonical model describing the delayed transformation from neural activity to BOLD signal.
- **HRF-weighted Context:** Token embeddings weighted according to their temporal contribution to the current TR via the HRF.
- **voxelwise Encoding Performance:** Correlation or related metric between predicted and measured voxel responses.
- **Cross-subject Generalization:** Testing whether models trained on one participant transfer to predicting neural responses of another participant.
- **Cross-stimulus Generalization:** Testing whether models trained on one dataset (e.g., Friends) predict neural responses for new stimuli (e.g., Movie10).

## STIMULI AND DATASETS

- **Friends Dataset (CNeuroMod):** A naturalistic fMRI dataset with over 50 hours of continuous viewing of the sitcom *Friends*.
- **Movie10 Dataset:** Approximately 10 hours of audiovisual film-based fMRI recordings used for out-of-distribution evaluation.
- **Held-out/Test Friends Season 3:** Episodes not included in training, used exclusively as an in-domain test set.
- **Transcript Input:** Time-aligned textual transcripts used to provide tokenized input to the LM.

## NLP AND MODEL ARCHITECTURE CONCEPTS

- **Transformer:** A neural architecture based on self-attention; GPT-2 and LLaMA-2 are transformer-based LMs.
- **LM Backbone:** The underlying pre-trained transformer parameters before adding brain-alignment modules.
- **LoRA (Low-Rank Adaptation):** A parameter-efficient fine-tuning technique using low-rank adapters injected into selected transformer layers.
- **Attention Projections (Q/K/V/O):** Query, key, value, and output linear projections within each self-attention mechanism.
- **Feed-forward Block (MLP Layer):** Two-layer fully connected network inside each transformer block.

## TRAINING MECHANISMS

- **Context Window (32 Tokens):** Number of preceding tokens considered when predicting BOLD responses for a given TR, chosen to match the HRF's temporal span.
- **Adaptive Loss Weighting:** Schedule that increases the brain-loss weight when LM perplexity plateaus, preventing representational collapse.
- **AdamW:** Optimizer combining Adam optimizer with decoupled weight decay.
- **Early Stopping / Validation Monitoring:** Stopping criterion based on validation loss plateauing to prevent overfitting.

EVALUATION METRICS AND PROCEDURES

- **voxelwise Correlation (r):** Pearson correlation between predicted and observed BOLD activity.

- **Ridge Regression / RidgeCV:** Linear regression with L2 regularization; RidgeCV selects voxelwise regularization strengths.

- **Permutation / Bootstrap Testing:** Statistical procedures for estimating voxel-level significance and controlling FDR.

- **VL-Commonsense (CoDa):** Benchmark evaluating visually grounded commonsense via noun–attribute predictions (e.g., color, shape, co-occurrence).

- **PMI Debiasing:** Pointwise Mutual Information correction used when scoring CoDa predictions to reduce lexical frequency bias.

ADDITIONAL CONCEPTS

- **Representational Collapse:** Degeneration of learned representations toward trivial or low-variance solutions during training.

- **Multisensory Inductive Bias:** Structure introduced by brain responses to audiovisual stimuli that enriches LM representations beyond text-only training.

- **Knowledge Distillation (Analogy):** Alignment process where the LM internalizes representational structure from brain activity, conceptually similar to distillation from a teacher model.

## A.1 FMRI PREPROCESSING

All BOLD data were preprocessed with `fMRIPrep` (Esteban et al., 2019). Preprocessing included skull-stripping, susceptibility distortion correction, motion correction, slice-timing correction, and co-registration of functional images to each participant's T1-weighted anatomy using boundary-based registration. The BOLD time series were resampled into each subject's native space, into standard MNI152NLin2009cAsym space, and onto cortical surfaces (fsaverage). Several confound regressors were extracted, including framewise displacement, DVARS, global signals, and physiological components (aCompCor and tCompCor). Motion outliers were annotated (FD $> 0.5$ mm or standardized DVARS $> 1.5$). All transformations were combined into a single interpolation step to minimize smoothing. For more information regarding the preprocessing steps, please refer to the CNeuroMod website `https://docs.cneuromod.ca/en/latest/DATASETS.html#friends`.

## A.2 EVALUATION ON VL-COMMONSENSE (CODA).

We implemented the CoDa multiple-choice downstream task by casting the items as conditional next-token scoring. Each example provided a noun $n$ and a relation $r$ (color/material/shape/size/co-occur), a candidate set $\mathcal{A}$, and gold weights $y$. For each model (GPT-2 base or +LoRA; LLaMA-2-7B base or +LoRA), we used a short prompt template in either declarative (e.g., "The typical color of a {noun} is {attr}.") or QA form (e.g., "Q: Typical color of {noun}? A: {attr}."). For each $a \in \mathcal{A}$ we computed the mean log-likelihood of the attribute *suffix* given the full prompt (leading space for GPT-2/BPE and raw string for LLaMA/SentencePiece). With PMI enabled, we subtract a simple answer-only prior averaged over prefixes like 'A:'/"Answer:": $\tilde{s}(a) = \log p(a \mid \text{prompt}) - \mathbb{E}_p[\log p(a \mid p)]$. Scores are softmax-normalized to $\hat{p}$; we report top-1 accuracy against positive golds and Spearman's $\rho$ with $y / \sum y$.

To control tokenization artefacts, we applied a per-candidate token capping $T$ via three policies at hand **both** (item kept only if all candidates are $\leq T$ under *both* tokenizers), **per-model** (filtered under the active tokenizer), or **off** (no filtering). The LoRA checkpoints are loaded on top of the frozen base model using standard PEFT modules on the matching base (PAD=EOS; no vocab resize unless required), and embeddings and vocabularies are never modified. It is important to indicate that througout the brain-fine-tuning, we did not introduce any text beyond the transcripts already paired with fMRI, and the only supervision signal was the voxel prediction objective. All reported results are micro-averaged across CoDa items.

### A.3 SPATIAL CONSIDERATIONS AND VOXELWISE ENCODING

Because voxel-level spatial organization varies substantially across individuals, using spatially structured encoders would require additional alignment and smoothing steps that introduce confounds into cross-subject generalization analysis. We therefore adopt a voxelwise linear encoder, consistent with standard naturalistic fMRI encoding approaches Wehbe et al. 2014; Huth et al. 2016; Caucheteux & King 2022; Schrimpf et al. 2021). This choice provides a direct and interpretable estimate of representational alignment while avoiding stronger spatial priors.

Across all LM variants, we extract the final-layer representations, as LoRA updates tend to primarily influence higher layers. These embeddings feed into a fully connected ridge-regularized encoder trained with an $L_2$ loss (no feature normalization in the encoding stack aside from the ridge normalization).

### A.4 ACROSS SUBJECT GENERALIZATION RIDGE REGRESSION MODELS

To evaluate the cross-subject generalization, we trained voxelwise ridge regression models `RidgeCV` that map the shared LM representation to each participant's voxel responses. We froze all LM and LoRA parameters after training on a source participant and evaluated the resulting representations on another subject. Because voxel locations and noise profiles differ across individuals, the encoding head cannot be transfered directly. Following standard practices in encoding model studies (e.g., Wehbe et al. 2015; Huth et al. 2016), we fit a new voxelwise RidgeCV decoder for each target subject. This post-hoc decoder does not modify the LM representations; it simply learns a subject-specific linear mapping from the shared representations space to each subject's voxel responses.

### A.5 DUAL TRAINING OBJECTIVE AND DYNAMIC WEIGHTING

In order to keep the balanced contribution of the language model properties as well as improve the brain encoding performances, we use a weighted sum of the brain loss and a language-modeling loss (token-level cross-entropy/perplexity) computed on the same inputs. Let $(w_{\text{brain}}, w_{\text{lm}})$ denote the weights on the two terms. During training we adapted these weights with a performance-triggered schedule: we initialized $w_{\text{brain}} = $ `brain_weight_start` and $w_{\text{lm}} = 1 - w_{\text{brain}}$, and increased $w_{\text{brain}}$ toward `brain_weight_final` whenever validation perplexity plateaued for `perp_patience` epochs (tolerance `perp_tol`), up to `sched_steps` steps after at least `sched_min_epochs` epochs. This preserves LM guidance early and prioritizes brain alignment as the LM loss saturates.

$$\mathcal{L} = w_{\text{brain}} \cdot \underbrace{\mathcal{L}_{\text{brain}}}_{\text{cosine similarity}} + w_{\text{lm}} \cdot \underbrace{\mathcal{L}_{\text{LM}}}_{\text{token CE / perplexity}} + \lambda \|W_{\text{ridge}}\|_2^2. \tag{1}$$

### A.6 OPTIMIZATION AND LEARNING RATES

For the brain-fine-tuned model, we used separate lr = `lr_finetune`, and weight decay = `weight_decay_LoRA` for the LoRA layers, while for the ridge group we used a smaller and fixed lr = `lr_ridge`, with no weight decay applied, given the shrinkage controlled by the $L_2$ penalty $\lambda = $ `l2_lambda` on the linear encoding layer. For the brain-from-scratch model, the transformer and linear layer were optimized jointly; when LoRA-style heads were not applicable, the backbone and linear layer were treated as separate groups with fixed learning rates.

We used gradient accumulation, automatic mixed precision for inference/validation, and early stopping with patience `patience` and improvement threshold `min_delta`. Batches without linguistic input were skipped to retain a meaningful loss signal; shuffling prevented leakage across splits. All encoding results are reported as voxelwise Pearson correlation coefficients ($r$) between predicted and observed BOLD time courses, without squaring or noise normalization; note that some prior work instead reports $R^2$ or noise-normalized scores, which are not directly comparable in magnitude.

Random shuffling of TRs ensured this had no impact on learning. (Note that our training, validation, and test data are separate and do not mix due to shuffling.) Test data remained unfiltered to preserve temporal continuity. Checkpoints were saved for improvements in the raw dual loss. This framework

enabled direct comparison of frozen, partially adapted, and fully retrained LMs under naturalistic stimuli.

### A.7 COMPUTE ENVIRONMENT

All experiments were conducted using a distributed mix of institutional and national high-performance computing (HPC) clusters. We used varying CUDA-compatible GPUs (e.g., V100, A100) with 1 GPU and 1 CPU core per job, and RAM allocations ranged from 60 GB to 150 GB based on the model size and data volume. Training jobs were parallelized across subjects, data sizes, and model configurations in a reproducible and modular structure via the $submitit$ Python module and $Hydra$ configuration system.

### A.8 PERFORMANCE EFFICIENCY

Given the scale of brain imaging data and associated GPU memory demands, we implemented a series of memory and computational optimizations to enable stable brain fine-tuning. LoRA was applied to attention and feedforward layers ($c_{proj}$, $c_{attn}$), updating only a small subset of parameters while freezing the pre-trained weights, which reduced memory usage and maintained generalization. This strategy allows efficient and stable parameter updates without modifying the full weight matrices, making it feasible to adapt large-scale models to brain data. LoRA preserves the majority of pre-trained parameters while providing sufficient flexibility for brain alignment, offering a practical compromise between frozen embeddings and full fine-tuning. During fine-tuning, only the LoRA parameters and the final fully connected layer were updated, enabling efficient adaptation to brain signals while maintaining computational tractability.

For the brain-fine-tuned models, we applied LoRA to a subset of the transformer blocks. For both models, the LoRA modules were inserted in the self-attention projections and the feed-forward projections of the last 6 layers for GPT-2 and the last 8 layers of the LLaMA-2-7B. We focused on the later layers, given that they carry more abstract and semantically rich representations that are most predictive of brain activity (Schrimpf et al. (2021)). This choice also limits the number of trainable parameters while allowing the LM backbone to adapt meaningfully to the brain prediction objective.

We utilized gradient checkpointing to recompute intermediate activations during backpropagation, thereby reducing memory usage. Gradient accumulation enabled small batch sizes without affecting training, while automatic mixed precision (AMP) via $torch.autocast()$ further cut memory by using float16. Large voxelwise fMRI batches were preloaded into CPU memory and moved to GPU as needed ($pin\_memory = True$), with $num\_workers = 0$ to avoid unnecessary GPU use. To prevent fragmentation and out-of-memory (OOM) errors, unused tensors were cleared manually ($torch.cuda.empty\_cache()$), validation ran without gradient tracking ($torch.no\_grad()$), and model caching was disabled ($use\_cache = False$). GPU memory was periodically monitored with $torch.cuda.memory\_allocated()$ and $torch.cuda.memory\_reserved()$. Together, these strategies enabled fine-tuning of high-dimensional brain-aligned language models with available GPUs.

### A.9 HYPERPARAMETER OPTIMIZATION

To identify optimal training configurations for the brain-aligned LM, we conducted systematic hyperparameter optimization (HPO) over the best-performing subject, focusing on learning rate, dropout rate, weight decay, and the L2 regularization strength (alpha) applied to the final linear encoding layer. The search space included learning rates of 1e-5, 5e-5, 1e-4, 5e-4, 1e-3, dropout rates of 0.0, 0.1, 0.2, 0.3, 0.5, weight decay values of 0.0, 1e-6, 1e-4, 1e-2, and ridge alpha values of 1e-5, 1e-4, 1e-3, 1e-2, 1e-1. Each configuration was trained for up to 30 epochs with early stopping based on validation loss (patience of 10 epochs), and the best model was retained for downstream evaluation. (See Figure S1)

Our experiments showed that dropout had minimal or negative effects on validation performance, while learning rate, weight decay, and ridge alpha were more influential. We attribute this to the low number of trainable parameters, which reduces overfitting risk and renders dropout unnecessary. Moreover, the stochasticity introduced by dropout likely destabilizes the learning signal in the presence of intrinsic brain activity. Based on these findings, we adopted a fully deterministic training strategy to improve stability, reproducibility, and interpretability across runs.

Table 2: Performance of GPT-2 Models Trained on Varying Brain Data Volumes

| Data | LR | DR | WD | Corr. | Val. Loss | PPL-F | PPL-M10 |
|------|------|------|---------|--------|-----------|-------|---------|
| 10h | 0.0001 | 0.1 | 0.0001 | 0.0330 | 0.9738 | 17.04 | 19.22 |
| 5h | 0.0001 | 0.5 | 0.00001 | 0.0217 | 0.9752 | 16.66 | 17.86 |
| 1h | 0.005 | 0.2 | 0.001 | 0.0086 | 0.9933 | 75.80 | 74.17 |

Table 3: The table summarizes model training hyperparameters and performance decided across all subjects based on a set of hyperparameter optimization experiments run with GPT-2 (124M) model. **Corr. LR** is learning rate, **DR** is dropout rate, **WD** is weight decay, **Corr.** is the mean voxelwise brain correlation; **Val. Loss** is the loss on held-out brain data. **PPL-F** and **PPL-M10** indicate model perplexity on Friends and Movie10 movie transcripts respectively.

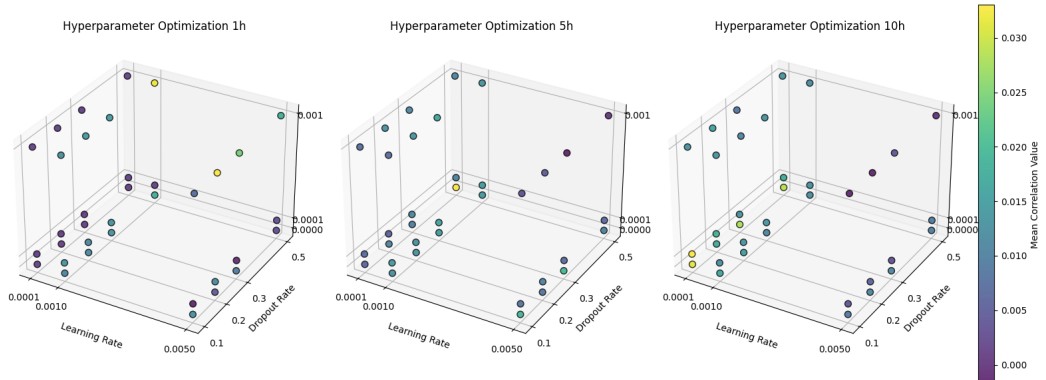

Supplementary Figure S1: **Visualization of hyperparameter optimization (HPO) across a 3D search space spanning learning rate, weight decay, and L2 regularization strength (alpha) for the brain-aligned model**: Each point in the cube represents a configuration evaluated during training, with color indicating validation performance. Shown here are the HPO results for the 1h, 5h, and 10h training conditions across subjects. Due to the observed plateau in hyperparameter sensitivity beyond 10h, we reused the 10h-optimized parameters for all larger training sets. Dropout was included in the initial search but was ultimately excluded due to its consistently neutral or negative effect on validation performance. The final selected hyperparameter values are listed in the accompanying table below.

For LoRA-based fine-tuning, we also evaluated the effect of using differing ranks $\{4, 8, 16, 32\}$. We found that ranks below 8 underfit brain alignment, while ranks above 32 yielded no meaningful gains besides increased memory requirements. We therefore adopted rank 8 for GPT-2 and rank 16 for LLaMA-2. Learning rates for LoRA parameters were constrained to the $10^{-5}$–$10^{-4}$ range to prevent destabilizing updates to the pre-trained backbone, whereas the linear encoding head benefited from slightly higher rates, consistent with prior-encoding model work. For the dual-loss regime, the weighting schedule was selected through small-scale sweeps on the validation set (0.1 to 0.9 with an increment of 0.1), with a dynamic schedule that gradually increased between stability and avoiding representational collapse. Overall, the final configuration reflected the settings that maximized the validation brain-prediction accuracy while maintaining stable updates to the LM backbone.

### A.10 OVERFITTING AND REGULARIZATION

Although the dataset is modest in terms of the number of participants (six), the naturalistic design yields a large number of training examples: 40 hours of Friends alone correspond to approximately 96k TRs per subject (1.49 s TR). To mitigate overfitting, our main brain-fine-tuned models use parameter-efficient LoRA adapters in a subset of layers plus a linear encoding head, combined with weight decay, early stopping based on validation loss, and a dual objective that maintains a language-modeling loss during training. We evaluate on held-out Friends episodes and on an out-of-

distribution movie dataset (Movie10) to directly measure generalization across stimuli and subjects. In contrast, the brain-from-scratch model is included as an ablation to probe the sufficiency of brain supervision alone and is not intended as a high-capacity, fully generalizable system.

### A.11 BOOTSTRAP-BASED STATISTICAL COMPARISON OF ENCODING PERFORMANCE

To assess the significance of the difference between two encoding model performances, we applied a paired-sample, block-wise bootstrap procedure. Let $s_{i,v}^{(m)}$ ($m \in \{1,2\}$, $i = 1, \ldots, N$, $v = 1, \ldots, V$) be the prediction score (e.g. Pearson $r$) of model $m$ on the $i$th validation sample at voxel $v$.

1. **Block-wise resampling.** Partition the $N$ validation samples into $M = \lceil N/B \rceil$ non-overlapping bins of size $B$ (we use $B = 10$). At each bootstrap iteration $k = 1, \ldots, K$ (with $K = 2000$), sample $M$ bins *with replacement* to form an index set $\mathcal{I}_k$, and then collect resampled scores $s_{i,v,k}^{(m)}$ for $i \in \mathcal{I}_k$.

2. **Difference matrix.** For each iteration $k$, compute the element-wise difference

$$D_{k,v} = \frac{1}{|\mathcal{I}_k|} \sum_{i \in \mathcal{I}_k} \big(s_{i,v,k}^{(1)} - s_{i,v,k}^{(2)}\big), \quad v = 1, \ldots, V.$$

and stack these into a bootstrap matrix $\mathbf{D} \in \mathbb{R}^{K \times V}$.

3. **95% confidence intervals.** For each voxel $v$, the lower and upper bounds of the two-sided 95% CI are given by the empirical percentiles:

$$\ell_v = \text{percentile}\big(\{D_{1,v}, \ldots, D_{K,v}\}, 2.5\big), \quad u_v = \text{percentile}\big(\{D_{1,v}, \ldots, D_{K,v}\}, 97.5\big).$$

4. **Voxel-wise $p$-values.** To test $\Delta_v = 0$ against $\Delta_v > 0$, compute $p_v = \frac{1}{K} \sum_{k=1}^{K} \mathbf{I}\{D_{k,v} \leq 0\}$.

We repeat this procedure for each pair of models, as well as between the training sizes of each model. We use the Benjamin Huxley False Discovery Rate (FDR) procedure to correct for multiple comparisons Benjamini & Hochberg (1995).

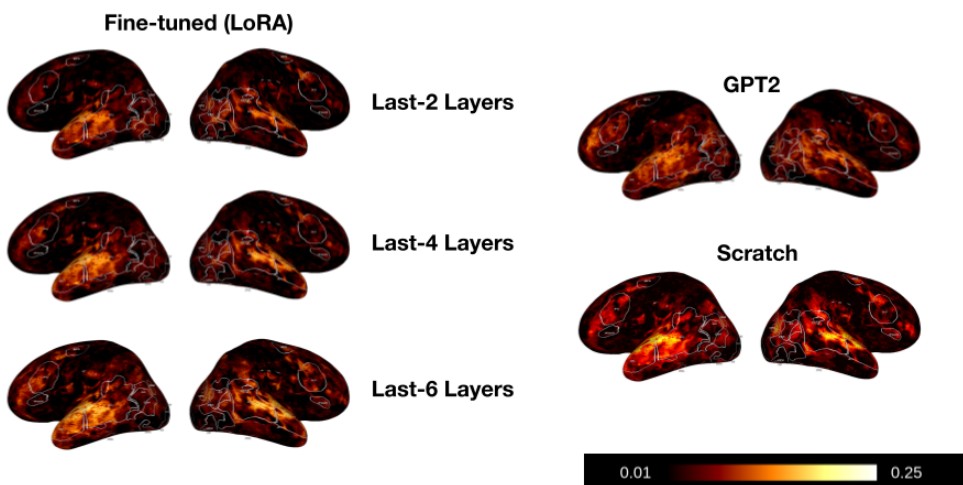

Supplementary Figure S2: **Selection of the Finetuning Layers:**We explored the impact of LoRA adapter placement depth by selectively applying adapters to the top 2, 4, and 6 transformer layers of the base model while keeping the rest of the network frozen. Each configuration was fine-tuned using the same downstream training setup. We observed that adapting the final 6 layers of GPT-2 consistently yielded the best performance across validation tasks, aligning with prior findings that mid-to-upper transformer layers capture task-relevant representations most effectively during parameter-efficient fine-tuning Li & Liang (2021), Hu et al. (2022). In a similar vein, we adapted LoRA with the last 8 layers of the LLaMA-2. Consequently, we adopt the configuration with LoRA modules attached to the selected top layers for all subsequent experiments reported in the main paper.

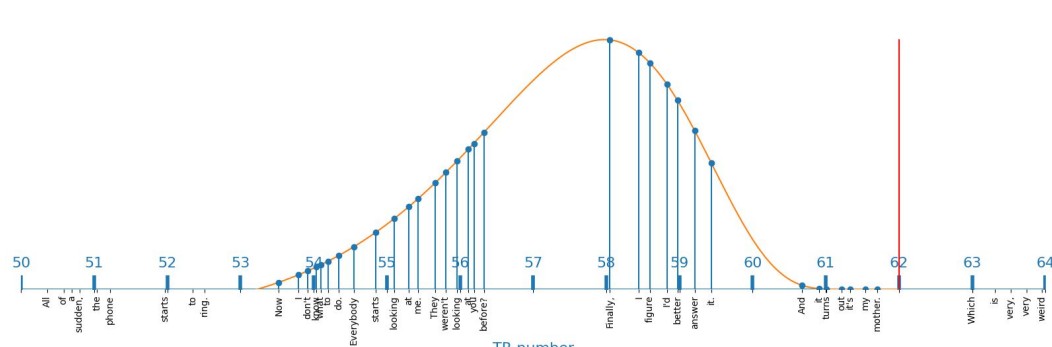

Supplementary Figure S3: **Illustration of HRF-weighted token integration for a single TR.** The plot shows the canonical single-gamma hemodynamic response function (HRF; orange curve) aligned to a target TR (red vertical line). Each blue tick marks a TR, and the blue stems show the HRF weight assigned to that token based on its temporal offset. For every TR, we select the 32 most recent tokens (covering roughly 5-6 seconds of preceding speech) and weight their final-layer LM embeddings by the HRF evaluated at the corresponding delays. The weighted embeddings are then linearly combined to produce the temporally aligned representation used for voxelwise BOLD prediction. This procedure implements a biologically motivated convolutional integration window of approximately 12-15 seconds, reflecting the sluggish temporal dynamics of the BOLD signal.

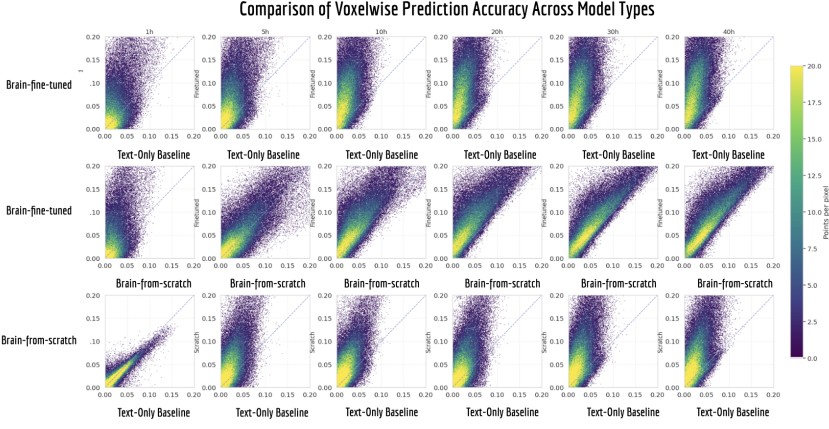

Supplementary Figure S4: **Comparison of voxelwise correlation values ($R$) for the brain-fine-tuned LLaMA-2 model and the text-only LLaMA-2 baseline across cortical regions.** Voxels were ranked by model performance within each region of interest (ROI) and only the top 10% are shown to highlight representational differences at ceiling. ROIs are grouped by functional domain (Language, Face/Body, Scene, Early Visual) and each boxplot reflects hundreds of voxels per model. The brain-fine-tuned model generally shows higher correlations in many language, scene, and early visual ROIs, whereas the text-only baseline achieves higher correlations in anterior and posterior temporal language regions (ATL, PTL). Significance brackets reflect Mann–Whitney U tests (asterisks indicating levels of significance, e.g., **** $p < 0.0001$), demonstrating reliable differences in voxelwise encoding between the two models across cortical systems.

Supplementary Figure S5: **Comparison of the brain-fine-tuned, brain-from-scratch and text-only baseline models voxel density.** Note that the text-only baseline here has the same architecture as the brain-fine-tuned network, but the entire language model base is frozen and only the last layer is trained).

### a) Brain-from-scratch Model Scaling Across Data Sizes

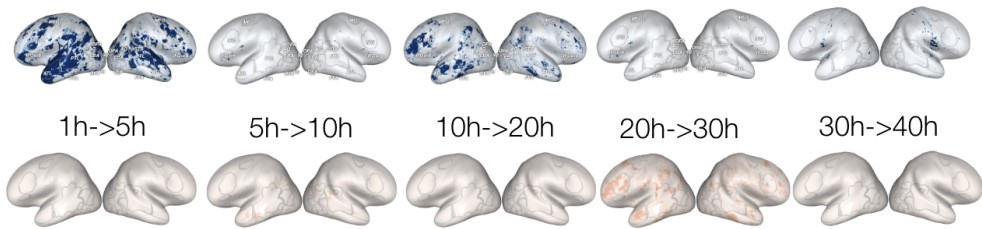

### b) Brain-fine-tuned Model Scaling Across Data Sizes

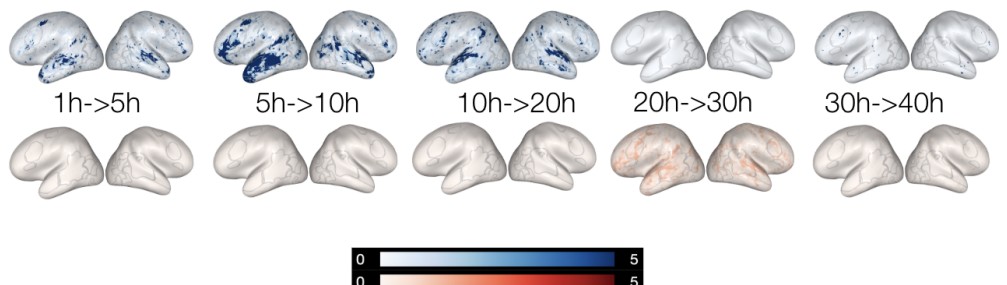

Supplementary Figure S6: **Data Size Differences across Brain-from-scratch and Brain-fine-tuned Model:** Warm-colored regions spotlight where each step up in training data brings measurable boosts in brain-encoding accuracy. While individual gains may be modest, their consistent appearance across increments highlights a clear trend: more data steadily unlocks new pockets of improved prediction. Cool-colored regions would flag any places where adding data backfired, but here they're effectively absent once corrected, confirming that larger datasets never erode performance and only serve to strengthen the model's ability to mirror neural responses. A similar trend was observed for the brain-from-scratch trained model, showing us that more data provides better generalizability to a certain degree, then staggers around a certain volume and repeats its performance levels from 20h onwards.

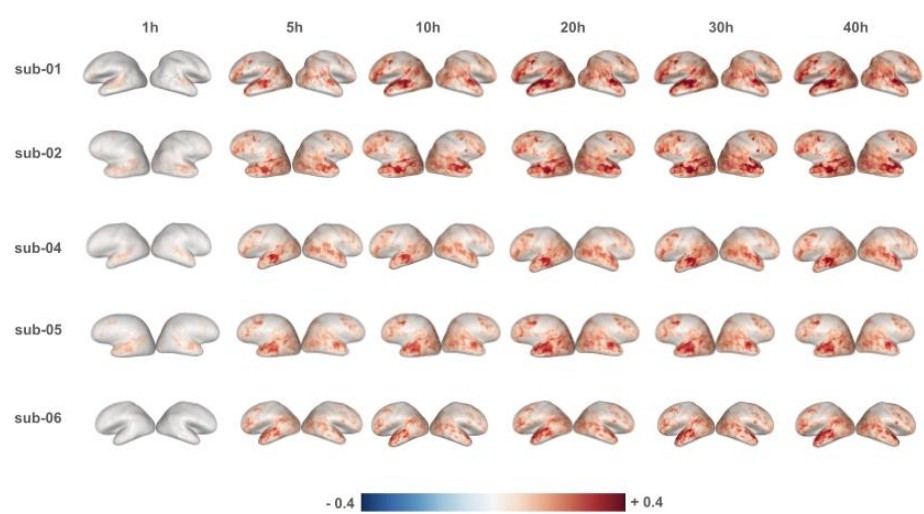

Supplementary Figure S7: **Across-subject brain-fine-tuned model data size scaling maps**: Predictive performance of the brain-fine-tuned model on brain maps of the five remaining subjects, shown across increasing amounts of training data. Each row corresponds to a subject, and columns reflect the model's fine-tuning with progressively larger subsets of the training set. Performance increases with more data, particularly in regions associated with language processing, plateauing around 20h onwards. These results illustrate how additional data improves model generalization and spatial coverage of accurate predictions.

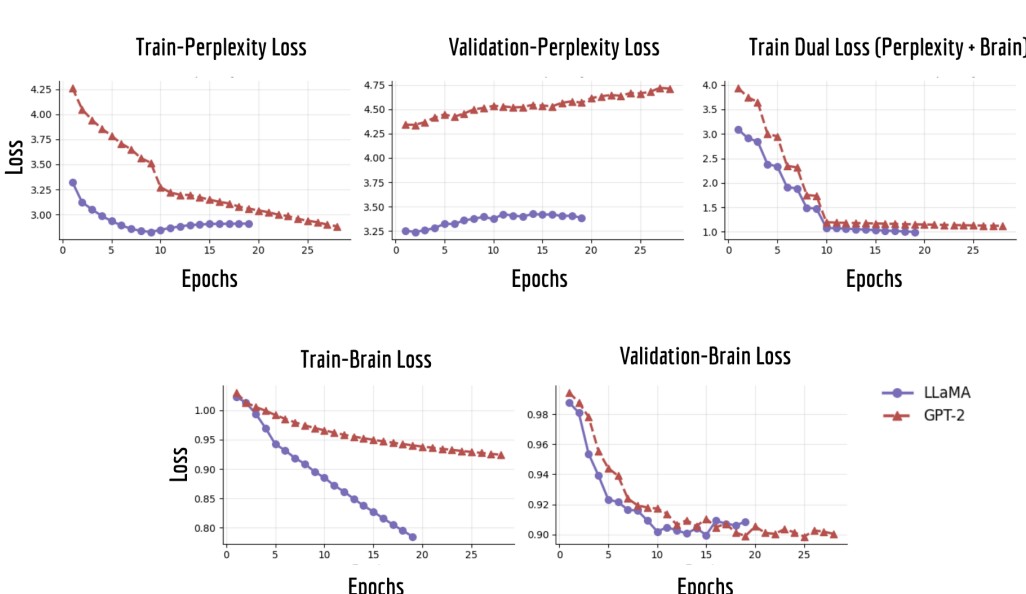

Supplementary Figure S8: **Dual-loss training dynamics for GPT-2 and LLaMA models aligned to fMRI responses.**: Training dynamics of dual-loss optimization for GPT-2 and LLaMA-2 fine-tuned to predict voxelwise fMRI responses (20h, sub-03). Each curve shows the evolution of the language objective (validation perplexity) and the neural objective (brain-prediction loss) across epochs. Early epochs are dominated by perplexity minimization, reflecting efficient token-level learning. As perplexity gains saturate, degradation of language performance. This coupling of objectives demonstrates that neural constraints can be integrated into the representational space of large language models without erasing linguistic knowledge. LLaMA-2 exhibits a more pronounced reduction in brain loss while maintaining stable perplexity, suggesting greater capacity for accommodating neural structure. The modest increase in the validation perplexity during later epochs also coincides with the scheduled shift in optimization weight from language prediction to neural alignment, reflecting the expected trade-off induced by the dual objective.

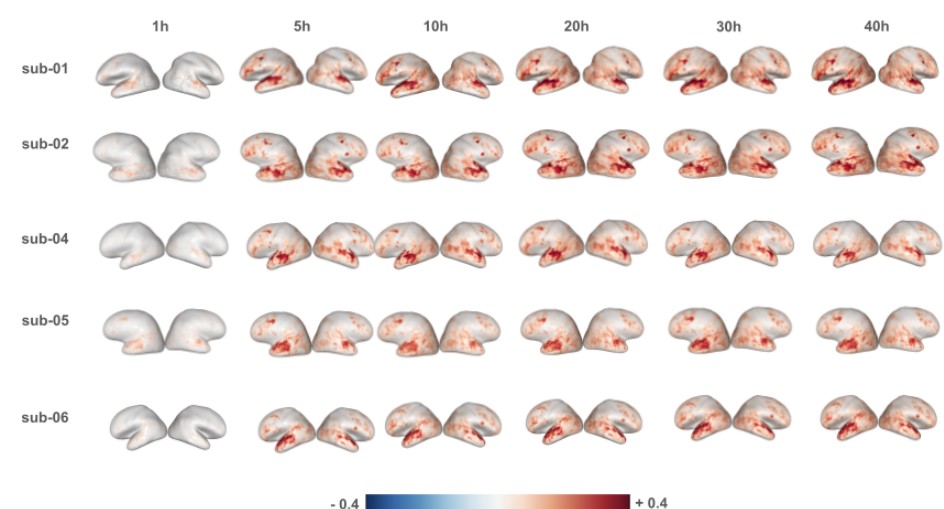

Supplementary Figure S9: **Across-subject brain-from-scratch model data size scaling maps**: Predictive performance of the brain-from-scratch (GPT-2) model on brain maps of the five remaining subjects, shown across increasing amounts of training data. Each row corresponds to a subject, and columns reflect the model's fine-tuning with progressively larger subsets of the training set. Performance increases with more data, particularly in regions associated with language processing, plateauing around 20h onwards. These results illustrate how additional data improves model generalization and spatial coverage of accurate predictions for the brain-from-scratch model, similarly to the brain-fine-tuned model.

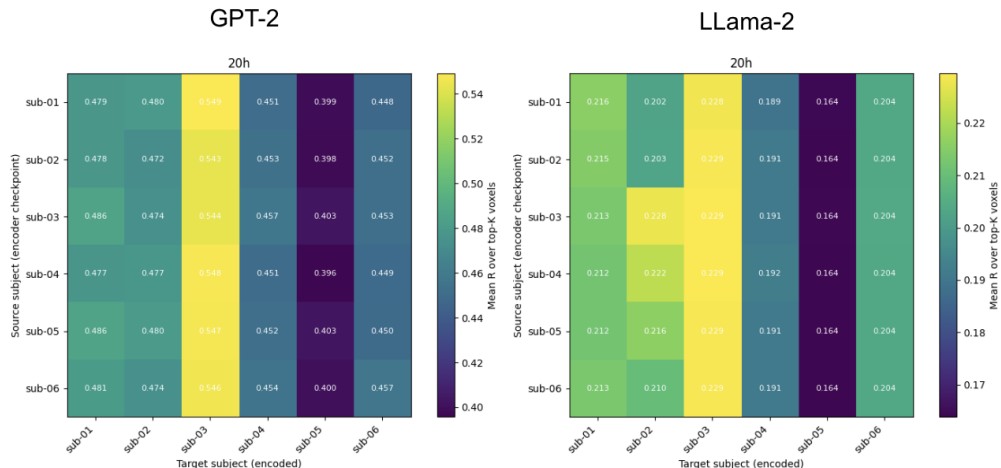

Supplementary Figure S10: **Cross-subject generalization**: The heat maps showing cross-subject encoding correlation maps for GPT-2 (left) and LLaMA (right) trained with 20h of the Friends dataset and tested on the left-over test dataset. Each cell shows the mean correlation over the top 2K voxels when models trained on one subject (rows) are evaluated on another subject's brain responses (columns). Performance remains robust across nearly all subject pairs, demonstrating that brain-informed supervision captures representational structure shared across individuals rather than overfitting to single-subject idiosyncrasies.

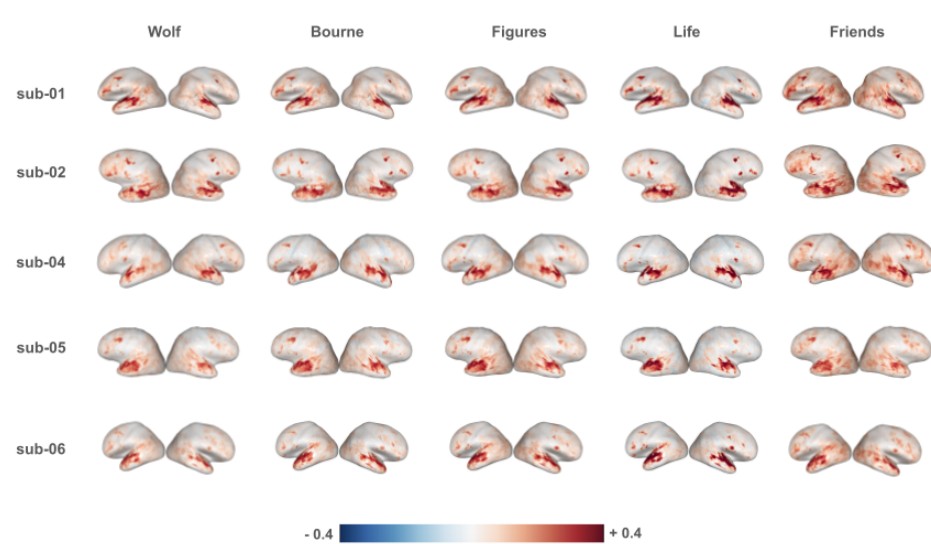

Supplementary Figure S11: **Across Subjects Brain-from-scratch Model Out-of-Sample Generalization**: Predictive performance of the brain-fine-tuned model visualized on brain maps for the five remaining subjects, evaluated on held-out (out-of-sample) movie types. These maps demonstrate the model's generalization ability to unseen stimulus domains and highlight consistent language-selective regions across subjects, despite variation in content.

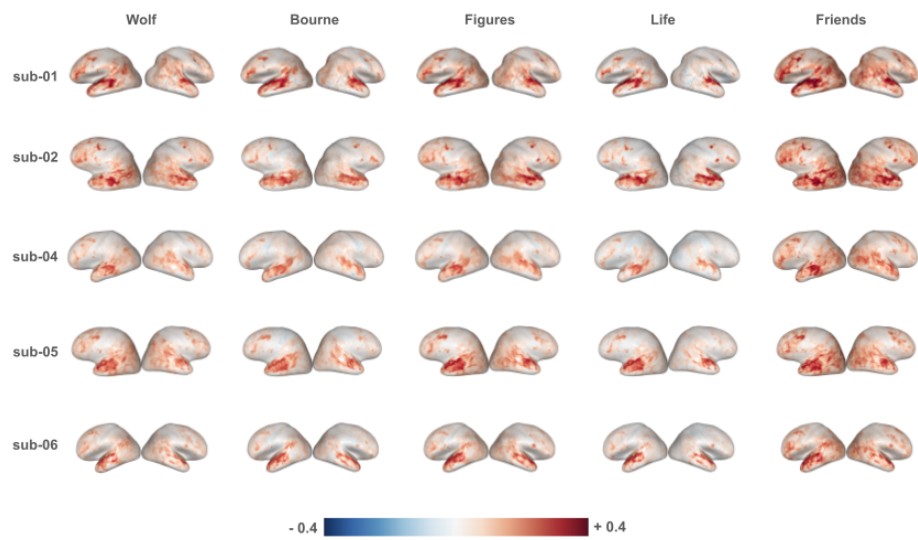

Supplementary Figure S12: **Across Subjects Brain-Fine-Tuned Model Out-of-Sample Generalization:** Predictive performance of the brain-fine-tuned model visualized on brain maps for the five remaining subjects, evaluated on held-out (out-of-sample) movie types. These maps demonstrate the model's generalization ability to unseen stimulus domains and highlight consistent language-selective regions across subjects, despite variation in content.

