# OpenReview forum: "Brain-Informed Language Model Training Enables Scalable and Generalizable Alignment with Human Brain Activity"
_ICLR.cc/2026/Conference — Submitted to ICLR 2026_

### Official Review · Reviewer_Skxm · 2025-10-30

**Soundness:** 2
**Presentation:** 2
**Contribution:** 3
**Rating:** 6
**Confidence:** 3

**Summary:**

This paper investigates how fMRI recordings can be used to fine-tune large language models (LLMs) toward human brain activity. The authors propose a dual-objective framework combining standard language modeling with brain alignment, leveraging over 50 hours of naturalistic movie-watching fMRI data. Experiments on GPT-2 and LLaMA-2 show consistent improvements in voxel-wise encoding, cross-subject generalization, and downstream visually grounded commonsense tasks. The study suggests that neural supervision can inject multimodal, human-like structure into text-only LMs.

**Strengths:**

* Interesting and novel idea, directly training LMs with human brain activity rather than merely testing alignment.
* Strong empirical design using large-scale naturalistic fMRI datasets with both within- and cross-subject validation.
* Careful exploration of scaling effects (data size and model size) and clear demonstration that alignment benefits grow with scale.
* Connection to downstream commonsense reasoning provides an interesting link between neuroscience and AI representation learning.

**Weaknesses:**

* The paper may inadvertently compromise the double-blind review process: Appendix A.3 explicitly names computing clusters and institutions (e.g., CMU, University of Montreal), which should be anonymized.
* Several figures (e.g., Fig. 2/3/5/6) are unclear or potentially misleading, the metrics are not labeled, making it difficult to interpret what is being compared or measured.
* The methodological description lacks precision. The "separate Ridge model" in Sec. 4.3 is not clearly explained, especially whether it was trained using all subjects or subject-specific data, and the strong cross-subject consistency in Appendix Fig. 12 is puzzling without further justification.
* The evaluation procedure on the CoDa dataset (Sec. 4.5) is underspecified; the paper does not clearly define how the metrics were computed.
* The paper focuses entirely on brain alignment and visually grounded reasoning but does not explore how brain fine-tuning affects performance on traditional text-only tasks. Understanding whether language quality is preserved or degraded is important for assessing the broader impact of this approach.

**Questions:**

See above.

---

> ### Author Response · Authors · 2025-12-03
> **Rebuttal by Authors to Reviewer Skxm**
>
> We thank to the reviewer for pointing to the crucial aspects in the paper that needs to be addressed. Please find below our rebuttal to the raised points.
>
> **Regarding anonymization**: We appreciate the reviewer for pointing out this very delicate detail. Now we have made the details anonymous.
>
>
> **Regarding figure updates and clarity in the ridge model**: We added a clearer description of the Ridge model used for the across-subject generalization to Appendix A.3, by defining its framework and usage. For cross-subject generalization, we train voxelwise ridge regression models, RidgeCV from sklearn, that map the shared LM representation to each participant’s voxel responses. We froze all LM and LoRA parameters after training on a source participant, extracting the representations  and evaluating these representations on another subject. Because voxel locations and noise profiles differ across individuals, the encoding head cannot be transferred directly. Hence, we fit a new voxelwise RidgeCV decoder for each target subject.
>
> **Regarding evaluation of the CoDA dataset**:  We thank the reviewer for this helpful suggestion. We added clarifications regarding the CoDa methodology in Appendix A.2 with more details regarding the exact prompt, format, scoring procedure, PMI style devising, tokenization controls, and LoRA loading details to make the whole process more transparent and reproducible.
>
> **Regarding preserving the language ability of LMs**: We thank the reviewer for giving us a chance to restate the scope of our work with this question. In this work, we did not aim to improve or modify the language or the performance of the language models that have been used in the backbone, and we intentionally made no claims that brain-fine-tuning enhances the model’s traditional NLP capabilities. The goal of this work is instead to use pre-trained LMs as representational backbones for modeling neural responses under naturalistic stimulation and to evaluate whether light, brain-supervised adaptation improves brain prediction and visually grounded reasoning, not linguistic competence. With these aims in mind, we did not conduct evaluations on standard text-only tasks. Especially in our model framework, we adopted a dual-loss regime to maintain the pretrained LM objective during fine-tuning, which is a known strategy for preventing catastrophic forgetting and preserving core capabilities (Caruana, 1997; Goodfellow et al., 2014; Aghajanyan et al., 2021). Additionally, the use of LoRA adapters has also been another assurance to keep the base model’s parameters largely unchanged, and adaptation to only occur in a constrained number of parameters to further limit any drift in the language backbone. We now emphasize in the revised version of the manuscript that our objective is representational alignment with brain data, rather than improving the underlying LM on traditional NLP benchmarks, and that brain fine-tuning is evaluated only in that context. To make this angle of our work clearer, we have added a declaration in both the Introduction and Methods - Objective and Dynamic Weighting section.
>
> Importantly, our training dynamics empirically confirm that this design successfully preserves language ability. As shown in Appendix Fig. 8, validation perplexity remains stable throughout training for both GPT-2 and LLaMA, even as the brain-prediction loss continues to decrease. Early epochs are dominated by improvements in perplexity, after which the dynamic weighting shifts emphasis toward neural alignment. Crucially, perplexity does not collapse once neural supervision takes effect, demonstrating that the dual-loss schedule maintains the pretrained language priors while enabling the model to integrate brain-derived structure. We have now added a short explanation of this behavior by referring the reader to the Appendix.
>
> We now highlight this point in the revised manuscript to make clear that our objective is representational alignment with brain data, not enhancement of linguistic performance, and that our training procedure is explicitly designed, and empirically show to preserve the underlying language capabilities of the pretrained models.

---

### Official Review · Reviewer_SFPh · 2025-10-31

**Soundness:** 2
**Presentation:** 2
**Contribution:** 2
**Rating:** 2
**Confidence:** 4

**Summary:**

The paper explores whether functional MRI (fMRI) recordings of human brain activity can serve as a supervisory signal to train large language models (LLMs) toward more human-like, multimodal representations. Building on over 50 hours of fMRI data from participants watching Friends and 10 hours of additional movie data, the authors fine-tune GPT-2 (124M) and LLaMA-2 (7B) using Low-Rank Adaptation (LoRA) within a dual-objective framework that balances standard language modeling loss with a brain alignment loss. Through systematic experiments, the authors show that brain-informed fine-tuning improves voxel-level encoding accuracy across auditory, temporal, and frontal cortical regions, scaling with both model size and training duration, and generalizes across participants and unseen movie stimuli.

**Strengths:**

1. The paper shows that brain activity can effectively guide LM training, not merely serve as a downstream encoding target.
2. The paper proposes a dual-loss optimization scheme dynamically balancing language and brain supervision to improve generalization and stability.
3. The experiments show that neural alignment captures shared cortical structures rather than individual noise.

**Weaknesses:**

1. Using fMRI to align LMs with brain activity is an incremental extension of prior work. For example, [1] has found that brain prediction performance scales logarithmically with model size from 125M to 30B parameter models. Additionally, understanding why or how brain alignment leads to better representations is missing. The argument that “brain signals provide multimodal inductive bias” remains intuitive but unquantified. The results show correlation gains but do not explain what representational dimensions change in the LM after brain fine-tuning. There is no layer-wise or feature-space analysis (e.g., probing semantic vs. perceptual dimensions, alignment metrics like CKA or RSA) to reveal how brain-aligned features differ from text-only ones. The reasoning of why the dual-objective training scheme prevents representational collapse is missing.

2. The “brain-from-scratch” and “text-only” models are included, but stronger baselines are missing.
3. The writing and the figure are hard to follow. For example, what do the blue and red mean in Figure 2?

[1] Antonello, Richard, Aditya Vaidya, and Alexander Huth. "Scaling laws for language encoding models in fMRI."

**Questions:**

1. Why can the proposed framework inject multisensory inductive biases? Can the authors quantify or demonstrate this claim?

2. Why does the dual-objective loss prevent representational collapse? Are there any insights?

3. The writing can be improved.

---

> ### Author Response · Authors · 2025-12-03
> **Rebuttal by Authors to Reviewer SFPh**
>
> We thank to the reviewer by their time and asking questions that helps with the manuscript conveying the message better. Here are our responses to the reviewers points raised.
>
>
> **Regarding understanding the benefit of brain-alignment**:  We respectfully disagree with the reviewer on this point. While prior work has indeed shown that frozen LM representations scale logarithmically in their ability to predict fMRI (e.g., Schrimpf et al., 2021; Caucheteux & King, 2022), our study asks a fundamentally different question: whether direct neural supervision changes a model’s representational space in ways that text-only scaling alone cannot? Unlike prior scaling papers, our work demonstrates that at fixed parameter count, brain fine-tuning yields large gains, while these gains improve cross-subject and cross-stimulus generalization, and brain supervision enhances perceptual and multimodal knowledge that do not emerge from text scaling alone.
> Thus, our study is orthogonal to scaling curves, while earlier work shows that larger models better approximate linguistic structure, we show that neural data provides qualitatively different supervision that reshapes representations in ways scaling alone does not produce.
>
> The argument that “brain signals provide multimodal inductive bias” remains intuitive but unquantified. The results show correlation gains but do not explain what representational dimensions change in the LM after brain fine-tuning.
> Why can the proposed framework inject multisensory inductive biases? Can the authors quantify or demonstrate this claim?
> Answer: We thank the reviewer for their question. The naturalistic movie viewing has been a useful tool in naturalistic neuroscience to be utilized in encoding research, showing its richness in multisensory, perceptual, associative, and contextual information that is not present in text-only corpora (Huth et al., 2026; Deniz et al., 2019; Lerner et al., 2011). The hypothesis is that when an LM is optimized to predict brain responses, the model is forced to encode the representational dimensions that are expected to support a broader perceptual-semantic space, which might not be truly accessible via a text-only trained model’s representation. This mechanism is already reflected in our results: Brain fine-tuning yields the largest gains in occipitotemporal and frontotemporal regions, which encode visual, object-semantic, and associative structure beyond linguistic features alone  (Huth et al., 2016; Binder & Desai, 2011). Additionally, the CoDa perceptual judgements experiments that we conducted showed brain-aligned models outperform text-only models, specifically on color, shape, and object co-occurrence attributes, which are dimensions known to be strongly grounded in neural representation and only weakly represented in text corpora.
>
> Taken together, these results suggest that brain supervision selectively amplifies perceptual and semantic dimensions that are underrepresented in text–only attributions. While we do not explicitly decompose the geometry of these representational changes, the combination of anatomically specific improvements in high-level perceptual and associative cortex, as well as behavioural gains on perceptual attributes, offers convergent support for the claim that brain signals act as a multimodal inductive bias. Indeed, more explicit quantification of the representational dimensions affected by brain-fine-tuning, whether it is early, mid, or high-level semantic features that shift with the features extracted from the brain data, would provide an important insight the underlying mechanism, and with the way that our pipeline was built, we are at a good position to investigate this aspect further in the future research as a complementary to our functional results. For the scope of this study, however, we focus on establishing the existence and behavioural/functional consequences of these representational changes as the initial step.

---

> ### Author Response · Authors · 2025-12-03
> **Rebuttal by Authors to Reviewer SFPh**
>
> **Regarding layer-wise and feature-space analysis suggestions**:  We agree that the layer-wise and feature space analyses offer valuable tools for understanding how model representations change; however, they were not within the scope of the present study, because we are using a single embedding (the language model’s output layer) and finetuning the entire network to predict brain activity. Thus the layerwise analysis is not as relevant. Our primary goal here was to establish whether large-scale naturalistic brain data improves language modeling performance and brain prediction accuracy, rather than to perform a mechanistic dissection of representational changes. That said, some of our results already hint towards the change in the representational reorganization: For example, we observe broadly distributed improvements across high-level language and semantic regions and shifts in voxel-wise prediction profiles compared to text-only models. The patterns we obtained are consistent with the idea that brain-aligned training emphasizes semantic and integrative features.
>
> **Regarding dual-objective loss prevent representational collapse**: We appreciate the reviewer’s question and have revised the manuscript to clarify why the dual objective (LM + brain loss) is necessary. The goal of this scheme is not to compensate for any lack of richness in the brain data, where our results even show that neural signals contribute complementary multimodal information, but rather to make sure that optimization remains stable while updating a fraction of its parameters with fMRI-derived features. We agree that our current experiments do not directly dissect how the dual objective shapes the internal representations of the language model, and we have revised the text to make this explicit. Our use of a joint language modeling and brain prediction loss is primarily a design choice motivated by existing machine learning theory. Concretely, the fMRI supervised part of training operates on a dataset that is much smaller than the text corpora used for pretraining, and the optimization is performed on a high-capacity model. In that regime, it is standard to retain the original pretraining objective, so that the model does not sacrifice its general linguistic competence while adapting to a new supervision signal. Prior work on multitask learning and continued adaptation shows that combining a strong pretrained objective with a new task tends to stabilize optimization and improve generalization by biasing the model toward shared structure instead of overfitting to the smaller task alone (Caruana, 1997; Goodfellow et al., 2014; Aghajanyan et al., 2021). We adopt the same principle here as the LM loss serves as a conservative prior that keeps the backbone close to its pretrained solution, while the brain loss encourages the emergence of features that better predict neural responses.
>
> Besides, LoRA provides an additional safeguard by restricting updates to a low-rank subspace (Hu et al., 2022), which is known to preserve most of the pretrained representations while allowing targeted adaptation. In our experiments, models trained with this dual objective maintain reasonable perplexity on held-out Friends and Movie10 text.

---

> ### Author Response · Authors · 2025-12-03
> **Rebuttal by Authors to Reviewer SFPh**
>
> **Regarding baselines and scratch model**:  As mentioned above, we have added supplementary plots that are quantitative and more clarity to Figure 5.  The notion of baseline here is complex. One is to add our hrf+linear layers and freeze the underlying language model (see results in the newly added Appendix Figure 5). Our brain-finetuned model greatly exceeds this baseline. Another baseline is to extract the features from a base language model and from the finetuned one, and train a separate ridge model. We show that in the regions closest to the auditory cortex/superior temporal gyrus, the ridge model has an advantage, but the finetuning benefits all the other, non-core language regions. As we argue in the paper, finetuning with brain data allows us to better model rich associations in areas outside the language cortex, which participate greatly in language comprehension (see also section 4.6).
>
> **Regarding figure readability**:  Thank you for pointing the missing metric was not included in the plot. Now we made the bar scale explicit with the indication that it shows Pearson's correlation and made the caption of the plot clearer regarding how to interpret the plot regarding the effect of training data volume on brain–LM alignment for GPT-2.
>
>
> **Regarding clarity in the writing**:  We have substantially revised the manuscript for readability and organization. Specifically, we improved the structure of the Results and Methods sections, added clearer explanations for the key methodological components, and rewrote several sections for conciseness and flow. We also redesigned some of the figures, and the axes and labels were made clearer, and edited their captions to better communicate our main takeaways. Together, these revisions improve the clarity and readability of the manuscript and convey the main hypotheses and supporting line of evidence provided throughout the experiments and results.

---

### Official Review · Reviewer_ME8m · 2025-10-31

**Soundness:** 3
**Presentation:** 1
**Contribution:** 2
**Rating:** 2
**Confidence:** 4

**Summary:**

This paper applies LLMs to predict fMRI voxel responses from text stimuli. The authors compare three training approaches: frozen LM, LoRA finetuning, and full finetuning across different LLMs (GPT-2, Llama, Mistral). They test cross-subject and cross-movie generalization on a movie-watching dataset.

While the results demonstrate that full finetuning with more data yields the best performance, the paper's contribution remains unclear. The architectural variants (LoRA/HRF) lack novelty, and the reported correlations are substantially lower than other LLM-based voxel prediction methods. The work would benefit from clearer positioning and substantial reframing to establish its distinct contribution to the field.

**Strengths:**

The paper provides detailed hyperparameter specifications that support reproducibility, and the clear section organization makes the methodology easy to follow. The experimental design includes valuable cross-subject and cross-movie generalization tests that are important for assessing model robustness in real-world applications.

**Weaknesses:**

### Architecture
The authors note that fMRI responses are "high-dimensional, have slow temporal dynamics, and are spatially structured," but the proposed architectures (frozen LM/LoRA/full finetuning) don't explicitly model these inductive biases. Missing architectural considerations include spatial voxel relationships, region-specific HRFs, or temporal dependencies from previous activations.

HRF details are missing: distribution type (Single Gamma/Canonical/Poisson), parameterization, and lag estimation methodology.

### Results quality
Mean correlations are very low and potentially insignificant without proper statistical testing that accounts for spatial structure. Other papers report order-of-magnitude higher results [1][2][3]. While datasets and preprocessing differ, the gap warrants detailed explanation.

### Presentation issues
- Figure 1: Links between panels A, B, C unclear; training regimes not referenced; text too small in B/C; technical terms (TR, HRF, BOLD) need explanation or simplification
- Figure 3: Color schemes unclear
- Inconsistent presentation throughout

### Typos
- line 250: "in 10" -> "in figure 10"
- line 329: "coincidentally" -> "fittingly"
- line 377: "Whether our model" -> "If our model"


[1] [On whether the relationship between large language models and brain activity is language-specific](https://2025.ccneuro.org/abstract_pdf/Gurel_2025_On_whether_relationship_large_language_models.pdf)
[2] [Hierarchical Brain–LLM Alignment Reveals Layer-Specific Neural Representations of Second Language Proficiency](https://www.biorxiv.org/content/10.1101/2025.06.17.660057v2.full.pdf)
[3] [fMRI predictors based on language models of increasing complexity recover brain left lateralization](https://arxiv.org/html/2405.17992v2)

**Questions:**

What is causing the anomaly in Figure 4 at 10h of training data with Llama?

How do you justify the low correlation results compared to other fMRI-LLM papers? What specific dataset or preprocessing differences account for the order-of-magnitude performance gap?

Can you provide detailed HRF specifications: distribution type, parameters, and lag estimation method?

Why don't the proposed architectures explicitly model the spatial and temporal structure of fMRI data that you highlight as important?

---

> ### Author Response · Authors · 2025-12-03
>
> We would like to thank to the reviewer for their time and help with the raising important points. Here are our answers to the reviewer listed below.
>
> ## Answers to the Weaknesses
> **Regarding the HRF model usage**: We thank the reviewer for raising these important points. Our goal in this work is to evaluate how much alignment between LMs and the brain can be learned without introducing additional architectural inductive biases. For this reason, we intentionally adopt the simplest and most widely used encoding framework, voxelwise linear modeling combined with canonical HRF, mirroring the past research (e.g., Huth et al., 2016; Caucheteux & King, 2022; Schrimpf et al.). This design allows us to isolate the contribution of the LM representations themselves, rather than the properties of the encoder architecture, and ensures direct comparability with prior work.
>
> Regarding the concerns raised on temporal structure, although our LM does not include an explicit temporal convolution, the HRF-weighting stage already introduces a biologically motivated temporal integration window. We now provide the full specification in the methods where we indicate clearly that we use a canonical single-gamma HRF with a peak at ~4-6 seconds, a physiologically realistic decay, and a temporal window of ~12-15 seconds. Token onset times are convolved with this HRF to produce the weighted LM feature at each TR. This approach aligns with standard practice in naturalistic fMRI Encoding and maintains temporal modeling consistency across subjects. In our cross-subject experiment, we additionally re-fit voxelwise RidgeCV regressions, which allows the linear decoder weights to capture voxel-dependent latency differences implicitly, a common approach when explicit per voxel HRF learning is not part of the model design.
>
> Regarding the concerns for the spatial structure, we agree that spatially structured encoders represent promising extensions. However, we also think that the voxel-wise modeling objective without the spatial constraints can also provide brain guidance to the model. More complex architectures require surface registration or anatomical alignment steps that would introduce additional confounds for the cross-subject generalization analysis that are central to our study. For this reason, we deliberately use a voxelwise linear encoder here. We revised the text accordingly and included all missing HRF details, distribution type, parametrization, temporal window, and lag handling in the Methods section.

---

> ### Author Response · Authors · 2025-12-03
> **Rebuttal by Authors to Reviewer ME8m**
>
> **Regarding the mean correlation values**: We believe the reviewer has a wrong impression of our result, perhaps due to a lack of clarity in our figures.  Our results do **not** present an order of magnitude difference with other work. In fact, we report a correlation of more than 0.4 and 0.5 in many brain regions. If it were the case that it was an order of magnitude lower than other work in the literature, that work would be reporting a correlation of 4 or 5, which, of course, is nonsensical. Perhaps the reviewer only looked at the numbers reported in figure 4 in which the numbers are between 0.01 and 0.02. These are averages **across the cortex**. Instead, the reviewer should look at the values of the correlation on the brain map with the provided legends, or the values in the scatter plots of Figure 3. Then it’s clear that we have a wide distribution of values, very competitive with state-of-the-art work, including the papers mentioned by the reviewer. Even if the reviewer focused on the average values in figure 4 these are comparable with the Bonnase-Gahot and Pallier paper figure 6 (the paper mentioned by the reviewer).
>
> Our paper reports raw voxelwise Pearson correlations (r ) across the entire cortex, with no noise normalization, no parcel averaging or spatial smoothing, or reliability filtering. We also predict single trial fMRI.  In contrast, the three cited papers report values that are not directly comparable to raw voxelwise r: Gurel et al. (2024) use ROI-averaged correlations on the Pereira dataset, which is an unusually high-SNR sentence-repetition design; they evaluate only language network ROIs, not full brain voxel distributions. The reported R values, although not indicated explicitly,  from a visual inspection seem to show a median of 0.3 across the language ROIs. When we perform the same analysis on the language ROIs, we get a median for some ROIs so be as high as 0.5, with the median language ROI median being around 0.3. We now add this to the paper in S4
>
>
> Kubo et al. (2025) report ROI-level prediction accuracy using a finite impulse response (FIR) model, with 5 temporal delays, significance masked voxels, and parcel averaging across 54 subjects listening to a controlled TOEFL passage. Their results are therefore inflated relative to raw voxelwise r. When translated into Pearson r units, their ROI level values fall in the 0.1 - 0.25 range, which is below the range we are observing.
>
>
> Bonnase-Gahot and Pallier (2024) analyze only the 25% most reliable voxels, resampled to 4 mm and often noise-normalized. Before filtering their full-brain voxelwise r distribution peaks around 0.1-0.18, they still remain below our value range. Also, in their Figure 6 their averages are of similar scale to ours.
>
>
> Our dataset and pipeline are deliberately conservative to maximize ecological validity, naturalistic audiovisual movies, no smoothing, no denoising, no voxel exclusion, and individual-subject encoding rather than group averages. Naturally, raw r-scores are lower under these structural conditions. Despite this, our best cortical regions reach r = 0.4 - 0.6, matching the top ranges reported in the cited works when expressed in the same metric.
>
>
> To avoid confusion, we have revised the manuscript to explicitly state that we report Pearson r (not a normalized metric), and clarify that raw voxelwise correlations cannot be compared numerically to ROI-averaged or noise-normalized scores.
>
>
> ### Potential issues raised by the reviewer
>
> **Regarding the Figure 1**: We thank the reviewer for this observation. We have now updated the experiment and model-related figure and made the figure caption more explanatory to address the mentioned details clearly in the text.
>
>
> **Regarding color schemes**:  We have updated the figure and the color scheme to better to be understood by the reader.
>
> **Regarding inconsistencies in presentations and typos**: We have fixed the typos as pointed and cleared the definitions and descriptions of the methods clearly throughout the main text and supported with the Appendix.

---

### Official Review · Reviewer_tQ6j · 2025-10-31

**Soundness:** 2
**Presentation:** 2
**Contribution:** 2
**Rating:** 2
**Confidence:** 4

**Summary:**

This paper explores how fMRI signals can be used not only as evaluation data / predictive objective to measure alignment with LLMs but also as supervisory signals to fine-tune them. The authors explore the potential for guiding LLM training with brain data. Authors test several strategies: 1. LoRA-based fine-tuning of pre-trained LLMs, 2. training LLMs from scratch using brain data, and (3) joint optimisation combining language and brain-alignment losses.

They report improvements in voxel-level encoding performance for brain-informed fine-tuning over brain-only models. They also highlight potential knowledge enhancement on visually grounded language benchmarks, suggesting that fine-tuning on fMRI data injects perceptual and associative priors that text-only training lacks, against text-only models.

**Strengths:**

The core idea of leveraging biologically grounded signals to inject multimodal capabilities into LLMs is original, well-motivated, and goes into the direction of more understandable AI models.

The use of the VL-Commonsense benchmark is a good idea to test whether brain supervision enhances perceptual knowledge (e.g., color, shape, co-occurrence).

The large-scale fMRI dataset (50h of Friends + 10h of movies) is valuable and significantly larger than prior studies.

**Weaknesses:**

Although the idea introduced is interesting and the motivation is clear, the paper in its current form is not yet ready for acceptance for several reasons.

1. First some parts of the methods are insufficiently explained:
- The concepts of brain-fine-tuned, brain-from-scratch, brain-only are not well explained early on in the paper (in the abstract and introduction) and complexify the understanding of the paper.
- The reasoning behind using an LLM architecture randomly initialised and train it only on brain data is not clear.
- The input data used (which texts? transcripts? captions?) is not described in the main text.
- Ridge modelling approach (4.3); is a new ridge model trained for each test subject? how does it impact the generalisation claim?

2. Some of the key hypotheses remain vague.
- Choice of some of hyperparameters, see next section for more details.
- The rationale behind training “brain-from-scratch” models is weak; given the data scale, the risk of overfitting is great.
- Missing or vague methodological justifications (e.g., why voxel-wise linear encoding? why not spatially constrained or surface-based models?.
- missing some real baseline to compare the performance of the model (simple correlation model between language embedding and brain activation to compare with fine-tuning).

3. The results section is not strong enough to convince about the value of the method and more importantly to support the main claims that are made throughout the paper ( "substantially improve", generalisation to new subjects, to new stimuli etc...).
- Some claims sound conclusive but are only suggestive, such as the generalisability to new stimuli and to new subjects. There are no quantitative metrics supporting the generalisation capabilities of the model to new subject and new stimuli. Only some figure are presented (in the appendix), without colour scale, and without any way of interpreting how good the performance are.
- Similarly the VL-Commonsense (CoDa) results would benefit from more detailed analysis (which objects are better classified etc.. ), what are the prompts, how differ semantically the answers of the models - as it is now, it is unclear what is actually improved/learned by the model by fine-tuning on brain data (is it the additional training on new input texts that is helping the model or is it the actual prediction task?)

4. The paper would benefit from better evaluation of the performance:
- Generally I found it very difficult to appreciate the results due to the absence of real baseline. How would perform a simple correlation model between language embeddings and fMRI activations (similar to J. Millet et al 2022 - Toward a realistic model of speech processing in the brain with self-supervised learning)
- Missing proper quantitative evaluation regarding generalisation performance, maybe some correlation scores per region? auditory, language area etc. Is the correlation score the best metric to assess the generalisation performance? what about stimuli (audio, visual or text) retrieval ? (see Dahan et al 2025 - SIM: Surface-based fMRI Analysis for Inter-Subject Multimodal Decoding from Movie-Watching Experiments)
- More systematic evaluations than correlation maps (difficult to read).

5. The paper overall needs substantial polishing in writing, structure, and presentation. The writing and figure design need substantial polishing for clarity and readability.
- Some of the concepts are not really introduced or cited (such as LoRA ).
- The phrasing is often unclear; several paragraphs (e.g., first in Related Work) would benefit from rewriting.
- Figure 1, should present the different training schemes more clearly.
- Legends in figures are not self-explanatory, it is not said what types of maps is shown what is show (Figure 6); Figure 4 lacks clarity.

**Questions:**

- Could you provide more rational of the use of 32 tokens as a context window?

- How are the LoRA layers selected within the model?

- How would you evaluate the risk of overfitting of training a LLM architecture on a relatively small dataset?

- What are the main limitations of the dataset (small N, inter-session variability)? How dependent are results of generalisation and sensory representation on this specific dataset? Have you considered richer benchmarks such as audiovisual captioning or narrative reasoning tasks to validate multimodal grounding?

- In terms of generalisation, what would be the comparative encoding scores when trained/tested directly on Movie10?

- it might be interesting as a lower bound of the experiment, to have the correlation score using a text-only model while finetuning only final layer to predict the brain activity (from text embeddings).

- It is not clear from the text to which training scheme refers the mention to "brain-only" ? I guess it is the brain-from-scratch but it is confusing.

---

> ### Author Response · Authors · 2025-12-03
> **Rebuttal by Authors to Reviewer tQ6j**
>
> We thank the reviewe for the detailed and thoughtful feedback. We are encouraged that you find the core idea original and well-motivated and appreciate your recognition of the value of our dataset and evaluation choice. Below we adress your concerns point-by-point.
>
> ### Answer to the Weaknesses:
>
> **Regarding model infrastructure distinction**: We appreciate the observation by the reviewer and agree that the model variants we have implemented in our work required clarity. In this revised manuscript, we now reference the training regimes better both in the Abstract but also in the Introduction, and describe them better in the Method section. We also added a shortened summary description in a glossary we added to Appendix A1 to help the readers navigate through the terminology we used.
>
>
> **Regarding random initialization of brain-from-scratch model**:  The brain-from-scratch model is not intended as a practical model, but as a conceptual ablation that tests whether linguistic or multimodal structure can emerge purely from brain supervision. This ablation isolates the contribution of the pre-trained LM, as well as allowing us to check for representational collapse, and evaluate how much prior linguistic structure is necessary for stable, generalizable brain alignment. We now clarify this motivation explicitly in Methods and Discussion and emphasize that the model is used as a lower-bound stress test rather than as a recommended approach.
>
> **Regarding input data**: We have added a clear description of the input text source in the Methods:  we use time-aligned transcripts of the audiovisual stimuli, which are derived from subtitle files manually, and aligned with the audio using the Assembly AI tool to obtain the time-stamps of each word in the discourse. Then, each word is tokenized, and the tokens belonging to a word are assigned the same time-stamp for pooling before HRF weighting.
>
>
> **Regarding ridge modeling approach**:  For cross-subject evaluation, we train a new ridge model per test subject, using that subject’s fMRI response, following standard practice in the encoding literature. Importantly, since the main idea of analysis is to see whether representations extracted from a model trained on a subject transfer to another subject, the ridge model we built does not modify the  LM parameters. Thus, it allows us to test generalization across subjects, and it also allows us to test whether training end-to-end has limitations vs. training a separate layer, as is sometimes the case in datasets with a moderate amount of samples. As we understand that this section is not clear in readers' minds, we now explain it explicitly in Section 4.3 and clarify how ridge regularization interacts with the fine-tuned features.
>
>
> **Regarding hypothesis definition:**  We have revised the Introduction to clearly articulate the three central hypotheses guiding our study to guide the reader and clarify the scientific expectations behind our experiments. Now we motivate that (i) Brain-informed fine-tuning should yield higher voxel-level encoding accuracy than text-only and brain-from-scratch baselines, (ii) brain-to-LM alignment should strengthen as both model size and the amount of fMRI supervision increase, (iii) brain-derived supervision should enrich the model’s representations with multisensory structure that text-only training underrepresents.
>
>
> **Regarding hyperparameter choices**: We have expanded the justification of hyperparameters in Appendix A.9 and summarized the key points in the main text, indicating that hyperparameters such as LoRA rank, learning rates, and loss schedule parameters were selected via grid search on the validation set.
>
> **Regarding voxelwise encoding**: We chose voxelwise linear encoding for two main reasons. First, it allows direct comparability with the large body of prior work on fMRI encoding models (e.g., Huth et al., 2016; Wehbe et al., 2014; Caucheteux & King, 2022; Schrimpf et al., 2021), where voxelwise linear mappings are the standard approach for assessing representational alignment with the brain. Secondly, spatial and geometric variability across subjects makes spatially structured encoders difficult to compare across participants, without additional registration pipelines, which would also confound our cross-subject generalization analysis. The goal of this work has been to isolate representational quality, not spatial regularization. We have added a clarifying text in the Methods explaining this methodological choice, and we now explicitly acknowledge in the Discussion that spatially structured encoders would be a valuable future work direction.

---

> ### Author Response · Authors · 2025-12-03
> **Rebuttal by Authors to Reviewer tQ6j**
>
> ### Answer to the Weaknesses
> **Regarding comparison to the baseline model**:  We have now added more quantitative comparisons with several models. See supplementary figures S4 for a direct baseline comparison. The notion of baseline here is complex. One is to add our hrf+linear layers and freeze the underlying language model (see results in the newly added supplementary figure 5). Our brain-finetuned model greatly exceeds this baseline. Another baseline is to extract the features from a base language model and from the finetuned one, and train a separate ridge model. We show in figure 5 (and in the new supplementary figures S4) that in the regions closest to the auditory cortex/superior temporal gyrus, the ridge model has an advantage, but the fine-tuning benefits all the other, non-core language regions. As we argue in the paper, fine-tuning with brain data allows us to better model rich associations in areas outside the language cortex, which participate greatly in language comprehension (see also section 4.6).
>
>
> **Regarding VL-Commonsense (CoDA) analysis**:  We added clarifications regarding the CoDa methodology in Appendix A.2 with more details regarding the exact prompt, format, scoring procedure, PMI style devising, tokenization controls, and LoRA loading details to make the whole process more transparent and reproducible. We also clarify a key point that  may not have been clear in the original submission: although our brain-fine-tuning uses a dual loss that includes a language modeling component, this stage does not expose the model to any new textual data beyond the \textit{Friends} transcripts already present during fMRI alignment. The language loss therefore acts only to preserve the pretrained linguistic structure rather than to introduce novel semantic associations or additional lexical knowledge. As a result, any improvements on CoDA cannot be attributed to extra language exposure, they instead arise from changes induced by the brain prediction objective, which reshapes the representational space toward perceptual and associative dimensions reflected in neural activity.
>
> To illustrate this effect, we now include a brief qualitative summary of what changes in the model’s predictions. The largest gains are observed in attributes such as color, shape, and co-occurrence, relations that require perceptual grounding and are typically underrepresented in text-only training. More abstract attributes show smaller effects, while a full feature space analysis is beyond the scope of this work, these category-specific trends provide initial evidence that brain-alignment encourages the model to encode sensory grounded associations rather than solely reinforcing textual co-occurrence statistics. We also added these revised clarifications to the main text.
>
>
> **Regarding generalization performance scores per ROIs**:  We thank to the reviewer for highlighting the need for more explicit quantitative evidence regarding regional generalization performance, In response, we have added an ROI level analysis to the Appendix S.4, reporting voxelwise correlation scores separately for language, face/body, scene selective and early visual cortices. These results show that brain-fine-tuning yields consistent improvements across most functional systems, with the strongest gains in frontal and temporal language areas (e.g. IFG and AG) and reliable yet smaller benefits in sensory regions such as V1 and V3. This analysis complements the whole-brain maps in the main text by quantifying how generalisation manifests across canonical cortical networks rather than only at the voxel level. In terms of stimulus decoding, we have found that for simple decoding (stimulus identification out of a set), the results are quite correlated with correlation. Given that and that correlation is standard in the field, we have opted for that metric.
>
>
> **Regarding polishing the writing**:  We have substantially revised the manuscript for readability and organization. Specifically, we improved the structure of the Results and Methods sections, added clearer explanations for the key methodological components, and rewrote several sections for conciseness and flow. We also redesigned some of the figures, and the axes and labels were made clearer, and edited their captions to better communicate our main takeaways. Together, these revisions improve the clarity and readability of the manuscript and convey the main hypotheses and supporting line of evidence  provided throughout the experiments and results.
>
> **Regarding clear consept descriptions**: We have added a declaration regarding why we adopted LoRA in the methods, as well as more details in the Appendix A.9 regarding the parameters that are chosen with integrating LoRA in the model infrastructure.

---

> ### Author Response · Authors · 2025-12-03
> **Rebuttal by Authors to Reviewer tQ6j**
>
> **Regarding Figure 1 model infrastructure depiction**:  The main focus of Figure 1 is to illustrate the generic pipeline that encompasses all possible infrastructures used throughout the study. Additionally, all model infrastructure details, and the parameters that have been used were explained thoroughly throughout the text and in the glossary to make the comparison for the reader better informed.
>
> ### Answer to the Questions:
> **Regarding the choice of 32 token-window size**: Our goal was to approximate the span of linguistic input that could contribute to a single fMRI TR, considering the slow nature of the hemodynamic response. With a TR of 1.49 seconds and a canonical HRF whose peak extends over several seconds, a window of 32 tokens typically corresponds to ~5-8 seconds of speech in our naturalistic stimuli, which covers the preceding context most likely to influence the BOLD response of that TR. To test this claim to hold true, we also experimented with other window ranges (16 vs 32 vs 64 tokens), where extending, for example, yielded negligible gains in predictive capability of the model, but increasing the compute and slightly destabilising the training. Therefore, 32 token-windows remained as a theoretically reasonable window given the stimuli and HRF. We clarified this rationale in the Methods and made the window size selection and its temporal coverage more explicitly reported in the text.
>
> **Regarding the selection of the LoRA layers**: In our implementation, we injected LoRA modules into the attention and feed-forward projections of the later transformer blocks, where prior work suggests the representations are most semantically aligned with brain responses. For GPT-2 this corresponded to the last 6 layers (half of the layers), while in LLaMA-2, we followed an analogous pattern and implemented LoRA to the last 8 layers. We chose this configuration based on validation performance and parameter-efficiency: concentrating adaptation in the upper layers allowed us to strongly influence the semantic repetitions that feed the encoding head while keeping the number of trainable parameters small. We have now described this configuration and rationale explicitly in Appendix A.8 Performance Efficiency section.
>
> **Regarding the evaluation of the overfitting and data size**:  We agree that overfitting is a central concern when adapting large architectures to neuroimaging datasets. We mitigate this risk in several ways. First, although we describe the dataset size in hours for readability, the effective number of training examples is large: 40 hours of Friends and additional Movie10 data corresponds to on the order of ~N TRs per subject and ~6N TRs across all six participants ( with N ≈ 90–100k), each with an associated context and voxelwise response. Second, our main brain-fine-tuned models do not update all LM parameters. Rather, we use parameter-efficient LoRA modules on a subset of layers and a linear encoding head, together with a dual objective that retains a language modeling loss. That setup keeps the backbone close to its pretrained solution and reduces the capacity of the brain-specific adapter, which empirically improves generalization. Third, we explicitly monitor overfitting through held-out data, which is a separate season (season 3) of Friends that is not used in training or validation, as well as we also use the Movie10 dataset to generalize this finding to. Encoding performance increases with training data and plateaus rather than diverging, and the learned patterns generalize across subjects and across movies, which is unlikely under severe overfitting. And finally,y we run various hyperparameter optimization experiments to find the most converging parameters for various parameter settings rather than assuming a priori values and running the experiments with. This also contributes towards more robust experimental settings as well as the utilization of the model’s predictive performance capability without overlooking overfitting. For the brain-from-scratch model, we agree that the overfitting risk is higher. We therefore treat it as a controlled ablation that isolates what can be learned from brain supervision alone rather than as a recommended practical model. We now clarify this role and discuss overfitting explicitly in Appendix A.10 Overfitting and regularization section,and add a short disclaimer in the limitation section.

---

> ### Author Response · Authors · 2025-12-03
> **Rebuttal by Authors to Reviewer tQ6j**
>
> **Regarding questions on any limitations of the dataset**:  In our study we benefit from the CNeuromod dataset, which prioritizes deep-phenotyping with extensive fMRI recordings from a small number of participants. We acknowledge that the limited number of subjects and potential inter-session variability may constrain population level conclusions. However our goal is not to estimate population averages but to investigate how neural supervision reshapes language model representations when exposed to prolonged, naturalistic brain signals from the same individual. For this purpose, dense longitudinal sampling, spanning tens of hours of continuous ecologically valid audiovisual stimuli, provides a level of representational stability and statistical power that is not attainable with large N but shallow datasets.
>
> Because the encoding task requires learning fine-grained mappings between model features and voxel responses, the deputy of per-subject sampling is more critical than the number of subjects. In fact, prior work shows that reliable vowelwise encoding, representational geometry estimation and neural alignment effects emerge only when models are trained on long-duration naturalistic stimuli with thousands of repeated observations per voxel. Our results demonstrate that the learned representations generalize across unseen movies and across subjects via linear remapping, suggesting that the observed sensory and multimodal benefits are not idiosyncratic to a single dataset or training regime.
>
> While we have not evaluated our models on audiovisual captioning or narrative reasoning benchmarks, yet, this is an exciting direction for future work. The present dataset already contains rich perceptual, linguist and semantic structure, enabling us to test whether neural supervision injects cognitively grounded priors into language representations. Incorporating external multimodal benchmarks would allow us to quantify how these neural priors translate into behavioral task driven improvements and we view such evaluation as a natural extension of this study rather than a prerequisite for validating the current claims.
>
> **Regarding generalization to the comparative encoding with Movie10**: We agree that training and testing directly on Movie10 would provide an in-domain reference point for that dataset. However, one of our primary goals in this work is to evaluate cross-dataset generalization of brain-aligned LMs, not to optimize performance on a small, specific movie set. This is the main reason we chose Friends as the training corpus and Movie10 strictly as an out-of-distribution test.
>
> Besides, Friends provides more than 50 hours of continuous audiovisual stimulation, whereas Movie10 comprises about 10hours. Given our large backbones, Movie10 alone is relatively small as a training source and substantially increases the risk of overfitting when used as the sole supervision signal, particularly for the 7B model. In contrast, Friends offers enough data to robustly fit the brain-alignment module and to meaningfully test how well this alignment transfers to new movies (movie10) that differ in pacing, cinematography, and narrative style. In other words, our design intentionally poses a harder and more informative generalization question. Can a model brain-aligned on one rich naturalistic domain, in this case Friends, predict brain responses in a different set of movies (movie10) it has never seen during training? Training and testing directly on Movie10 would mainly assess in-domain performance on a smaller dataset and would likely provide a modest upper bound for that specific stimulus set, but it would not address this cross-dataset transfer. Given space and compute constraints, we therefore prioritized the Friends on Movie10 regime as a cleaner test of generalization, and now clarify this rationale in the revised manuscript, but we fully agree it is a valuable direction to investigate.

---

> > ### Author Response · Authors · 2025-12-03
> > **Rebuttal by Authors to Reviewer tQ6j**
> >
> > **Regarding the brain-only reference in the text**:  Thank you for pointing out this ambiguity. In the original draft, we used “brain-only” in two senses, sometimes to mean a model trained with only a brain-prediction loss (no language modeling loss) and sometimes to refer to the randomly initialized “brain-from-scratch” model. We agree that this is confusing. In the revised manuscript, we standardize the terminology across the main text by using “brain-only” to refer to the training objective consisting only of the brain-prediction loss; no language modeling term is used, while “brain-from-scratch” model is consistently used to refer to the model that is a randomly initialized transformer LM trained solely on brain data. We also add a glossary in the Appendix A.1 to help the reader navigate the terminology used in the text.

---

### Meta-Review · Area_Chair_6WwX · 2026-01-12

**Summary:**

This paper investigates whether fMRI signals can be used as supervisory signals to fine-tune LLMs, proposing a dual-objective framework combining language modeling and brain-alignment losses. Across several LMs and naturalistic movie-watching datasets, the authors report improved voxel-wise encoding accuracy, some cross-subject generalization, and gains on visually grounded commonsense benchmarks.

The work is motivated and uses a valuable large-scale fMRI dataset, and reviewers note the careful exploration of scaling effects and cross-subject testing (tQ6j, Skxm). However, the overall contribution is limited and insufficiently substantiated. Several reviewers highlight that the core approach—LLM fine-tuning with fMRI supervision—is incremental relative to prior alignment and scaling studies, with limited architectural or conceptual novelty (ME8m, SFPh). Methodological choices are often weakly justified or unclear, including the role of “brain-from-scratch” models, HRF specification, ridge decoding for generalization, and missing or underspecified baselines (tQ6j, ME8m). Crucially, many central claims—such as multimodal inductive bias injection and generalization—remain only suggestive, lacking rigorous quantitative or representational analyses (SFPh). Presentation and clarity issues further hinder evaluation.

**Reviewer Concerns:**

Methodological Vagueness: Key training regimes, such as "brain-from-scratch," lacked clear justification, and the risk of overfitting on relatively small neuroimaging datasets was not sufficiently addressed (Reviewer tQ6j).
Insufficient Evaluation: The paper lacks strong baselines, making it difficult to assess the actual value of brain-informed fine-tuning over standard frozen embeddings. Furthermore, claims regarding generalization to new subjects and stimuli lacked rigorous quantitative support (Reviewer tQ6j, ME8m).
Presentation Issues: Figures lacked essential labels and scales, and the manuscript required significant polishing for clarity (Reviewer SFPh, Skxm).
Policy Concerns: Anonymity was compromised by naming specific institutions in the appendix (Reviewer Skxm).
Overall, while the rebuttal clarified several technical details, the evidence for the claimed multimodal benefits remains suggestive rather than conclusive.

**Reviewer Scores:**

It is very hard to tell. The authors did provide adequate answers to many of the questions.

---

### Decision · Program_Chairs · 2026-01-26

Reject